# BioE3 identifies specific substrates of ubiquitin E3 ligases

Orhi Barroso-Gomila [1,8], Laura Merino-Cacho [1,8], Veronica Muratore[1], Coralia Perez [1], Vincenzo Taibi[2], Elena Maspero [2], Mikel Azkargorta[1,3], Ibon Iloro[1,3], Fredrik Trulsson[4], Alfred C. O. Vertegaal[4], Ugo Mayor [5,6], Felix Elortza [1,3], Simona Polo [2,7], Rosa Barrio [1] ✉ & James D. Sutherland [1] ✉

Hundreds of E3 ligases play a critical role in recognizing specific substrates for modification by ubiquitin (Ub). Separating genuine targets of E3s from E3-interactors remains a challenge. We present BioE3, a powerful approach for matching substrates to Ub E3 ligases of interest. Using BirA-E3 ligase fusions and bioUb, site-specific biotinylation of Ub-modified substrates of particular E3s facilitates proteomic identification. We show that BioE3 identifies both known and new targets of two RING-type E3 ligases: RNF4 (DNA damage response, PML bodies), and MIB1 (endocytosis, autophagy, centrosome dynamics). Versatile BioE3 identifies targets of an organelle-specific E3 (MARCH5) and a relatively uncharacterized E3 (RNF214). Furthermore, BioE3 works with NEDD4, a HECT-type E3, identifying new targets linked to vesicular trafficking. BioE3 detects altered specificity in response to chemicals, opening avenues for targeted protein degradation, and may be applicable for other Ub-likes (UbLs, e.g., SUMO) and E3 types. BioE3 applications shed light on cellular regulation by the complex UbL network.

Protein ubiquitination is conserved in all eukaryotes and plays crucial roles in almost all cellular processes. Ubiquitin (Ub) conjugation is coordinated by a three-step enzymatic cascade, which can be reversed by the action of deubiquitinating enzymes (DUBs). This cycle is conserved among the different ubiquitin-like proteins (UbLs), each using their own set of enzymes, often depicted as E1 (activating), E2 (conjugating), E3 (ligating), and DUBs. Specificity of ubiquitin toward particular targets is achieved as the cycle progresses. In humans, two Ub E1 enzymes, around 40 E2s and about 700 E3 ligases cooperate to selectively target thousands of substrates[1]. The question of how substrate specificity is achieved might benefit from a compendium of targets for specific E3 ligases.

Ub E3 ligases are subdivided into categories, according to shared domains and modes of action for substrate modification[2]. The main family covers more than 600 RING (Really Interesting New Gene) type Ub E3 ligases. The RING domain allows the direct transfer of Ub from the E2 to the target protein by placing them in close proximity[3]. To function, some RING E3 ligases (e.g., RNF4; RING Finger protein 4), dimerize through their RING domain[4], or create multi-subunit complexes, (e.g., Cullin RING Ligases). CRLs can recognize diverse targets with specificity by forming complexes with >300 different substrate receptors[2].

In the case of HECT (Homology to E6AP C Terminus) and RBR (RING-Between-RING) E3 ligases, a covalent E3-Ub thioester

[1]Center for Cooperative Research in Biosciences (CIC bioGUNE), Basque Research and Technology Alliance (BRTA), Bizkaia Technology Park, Building 801A, 48160 Derio, Spain. [2]IFOM ETS, The AIRC Institute of Molecular Oncology, Milan, Italy. [3]CIBERehd, Instituto de Salud Carlos III, C/ Monforte de Lemos 3-5, Pabellón 11, Planta 0, 28029 Madrid, Spain. [4]Cell and Chemical Biology, Leiden University Medical Center (LUMC), 2333 ZA Leiden, The Netherlands. [5]Ikerbasque, Basque Foundation for Science, 48011 Bilbao, Spain. [6]Biochemistry and Molecular Biology Department, University of the Basque Country (UPV/EHU), E-48940 Leioa, Spain. [7]Dipartimento di oncologia ed emato-oncologia, Università degli Studi di Milano, Milan, Italy. [8]These authors contributed equally: Orhi Barroso-Gomila, Laura Merino-Cacho. ✉e-mail: rbarrio@cicbiogune.es; jsutherland@cicbiogune.es

intermediate is formed before passing the Ub to the recruited substrate. HECT type E3s present a conserved C-terminal HECT domain, which contains the catalytic cysteine for Ub conjugation and transfer[5]. There are 28 human HECT E3s, with diversity in their N-terminal substrate-binding and regulatory domains[6].

Ub modifications by E3 ligases are dynamic, spatial-specific, and often scarce in cells. Characterizing these events in vivo requires efficient and specific enrichment protocols to identify targets. Use of biotin-avidin technology[7] is used by molecular cell biologists in diverse molecular contexts, including ubiquitination (reviewed in[8]). The pairing of BirA, a biotin ligase from *E. coli*, and the AviTag, a minimal peptide substrate specifically modified by BirA[9], has been widely used to achieve site-specific biotinylation for in vitro and in vivo applications. Once biotinylated, an AviTag fusion protein can be purified using streptavidin, via tight binding and stringent washing[9]. For example, AviTag-UbL fusions (bioUbLs) co-expressed with BirA are specifically biotinylated, incorporating into targets in vivo, allowing their purification and identification using liquid chromatography-mass spectrometry (LC-MS)[10].

Although structural biology has improved our understanding on how E3 ligases work, the identification of substrates for a given E3 and discriminating between non-covalent interactors versus bona fide targets remains challenging. Various strategies have been employed, included some that bring together E3 ligases and UbLs by fusion or affinity, to enrich potential substrates[11–14]. Here we present BioE3, an innovative strategy designed to identify specific substrates of RING and HECT E3 ligases. By combining site-specific biotinylation of bioUbL-modified substrates with BirA-E3 ligase fusion proteins under optimized conditions, we demonstrate that BioE3 can be applied to Ub and SUMO E3 ligases. BioE3 specifically identified known and novel targets of RNF4 and MIB1, two RING-type E3s. BioE3 was further applied to additional RING E3s: a membrane-bound mitochondrial E3 (MARCH5) and a poorly characterized cytoplasmic E3 (RNF214), yielding novel targets that give insight into the biological roles of these enzymes. Lastly, we show that BioE3 can be adapted for HECT type E3 ligases, identifying known and novel targets of NEDD4. As many E3 ligases remain uncharacterized, BioE3 can potentially shed light on specificity, redundancy, and network interconnectivity regulated by cellular UbL modifications.

## Results

### BioE3: a strategy to label, isolate, and identify bona fide targets of E3 ligases

Determining the specific substrates for an E3 of interest is a crucial but challenging task that requires the development of new techniques. We postulated that the fusion of the biotin ligase BirA to an E3 ligase of interest, combined with the bioUbL strategy[15], could be used to identify specific substrates of E3 ligases, a method that we have named BioE3 (Fig. 1). Various optimizations improved the technique, as detailed in the following section. Briefly, BioE3 employs a version of

AviTag with lower affinity for BirA (called here bio[GEF], see below for explanation) fused to a UbL encoding gene. The bio[GEF]Ub is incorporated into a doxycycline-inducible lentiviral vector for generation of stable cell lines (HEK293FT, U2OS). BirA is fused to the E3 ligase of interest, which is then introduced into bio[GEF]Ub cells, previously grown in medium with dialyzed, biotin-depleted serum. DOX induction over 24 h leads to production and incorporation of bio[GEF]Ub into cellular substrates, with concomitant increase in BirA-E3 expression. Finally, exogenous biotin is added, allowing time-limited, proximity-dependent labeling of bio[GEF]Ub as it is incorporated by the BirA-E3 fusion onto specific substrates. This facilitates streptavidin capture of tagged substrates and identification by LC-MS.

### Engineering BioE3 specificity

The widely-used wild-type (WT) AviTag (hereafter called bio[WHE]) is optimized for efficient biotinylation and has high affinity for BirA, so we wondered how this would affect the ability to use the BirA-bio[WHE] pairing for detecting a transient proximity-dependent event like protein ubiquitination. To evaluate the spatial-specificity, we fused the bio[WHE] tag to a version of Ub that is not processable by DUBs (Ubnc; nc = non-cleavable, L73P mutation)[16], to reduce any recycling of biotinylated bio[WHE]Ub to sites other than where BirA is found. When bio[WHE]Ubnc was expressed together with BirA alone or a centrosome-targeted BirA (CEP120-BirA), we observed that the biotinylation of bio[WHE]Ubncs was general and unspecific, independently of the subcellular localization of BirA (see Supplementary Note 1 and Supplementary Fig. 1a–c). AviTag versions with lower affinity for BirA have been described[17,18] including one where the C-terminal WHE sequence is mutated to GEF (hereafter called bio[GEF]; Fig. 2a), and these mutants enhance proximity-dependent site-specific biotinylation. We compared bio[WHE]Ubnc and bio[GEF]Ubnc for levels of non-specific labeling by transfecting them into a BirA-expressing 293FT stable cell line (Fig. 2a). To control biotin labeling timings, cells were preincubated in biotin-depleted media prior to transfections and DOX induction (see Supplementary Note 1). Commercial AviTag antibody still detects bio[GEF] tag, despite the mutations, and bio[GEF]Ubncs are efficiently incorporated into substrates (Fig. 2a). As expected, non-specific biotinylation of bio[WHE]Ubncs was observed at both 0.5 and 2 h of biotin labeling, while bio[GEF]Ubncs showed no labeling (Fig. 2a). We also compared bio[WHE] and bio[GEF] in the context of SUMO1nc and SUMO2nc (containing Q94P and Q90P mutations, respectively, to avoid recycling by SENPs[19]) and observed similar results, that is general labeling of bio[WHE]-SUMO1nc and SUMO2nc, but no labeling for bio[GEF] counterparts (Fig. 2b). Thus, using bio[GEF] and controlling biotin availability and timing, non-specific labeling by BirA can be avoided, thereby enabling the BioE3 strategy.

To test the BioE3 method, we expressed fusion proteins of BirA together with RNF4[12] or MIB1[20], two well-characterized RING type Ubiquitin E3 ligases, in biotin-depleted U2OS TRIPZ-bio[GEF]Ubnc or bio[WHE]Ubnc cells, followed by 2 h of biotin-labeling (Fig. 2c). BirA was

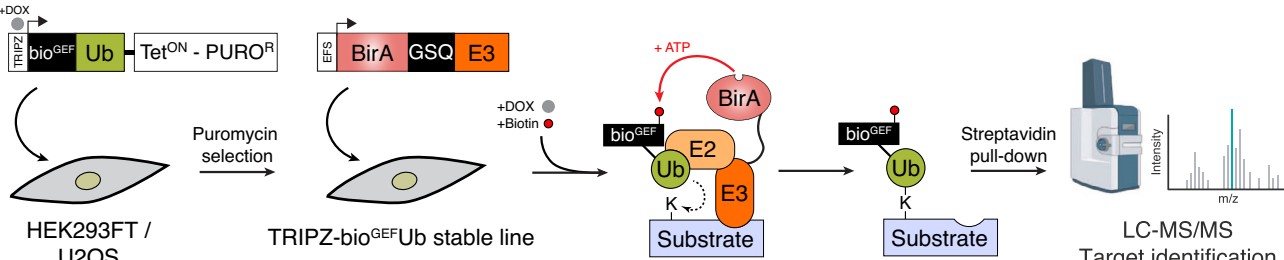

**Fig. 1 | Identification of substrates of E3 ligases: the BioE3 strategy.** Schematic representation of the BioE3 strategy and the constructs used. bio[GEF], low-affinity AviTag (see text); DOX, doxycycline; EFS, elongation factor 1α short promoter; GSQ, Gly-Ser-Gln flexi-rigid linker; PURO[R], puromycin resistant cassette; Tet[ON], tetracycline inducible promoter; TRIPZ, all-in-one inducible lentiviral vector; Ub, Ubiquitin. Some elements sourced from BioRender.com.

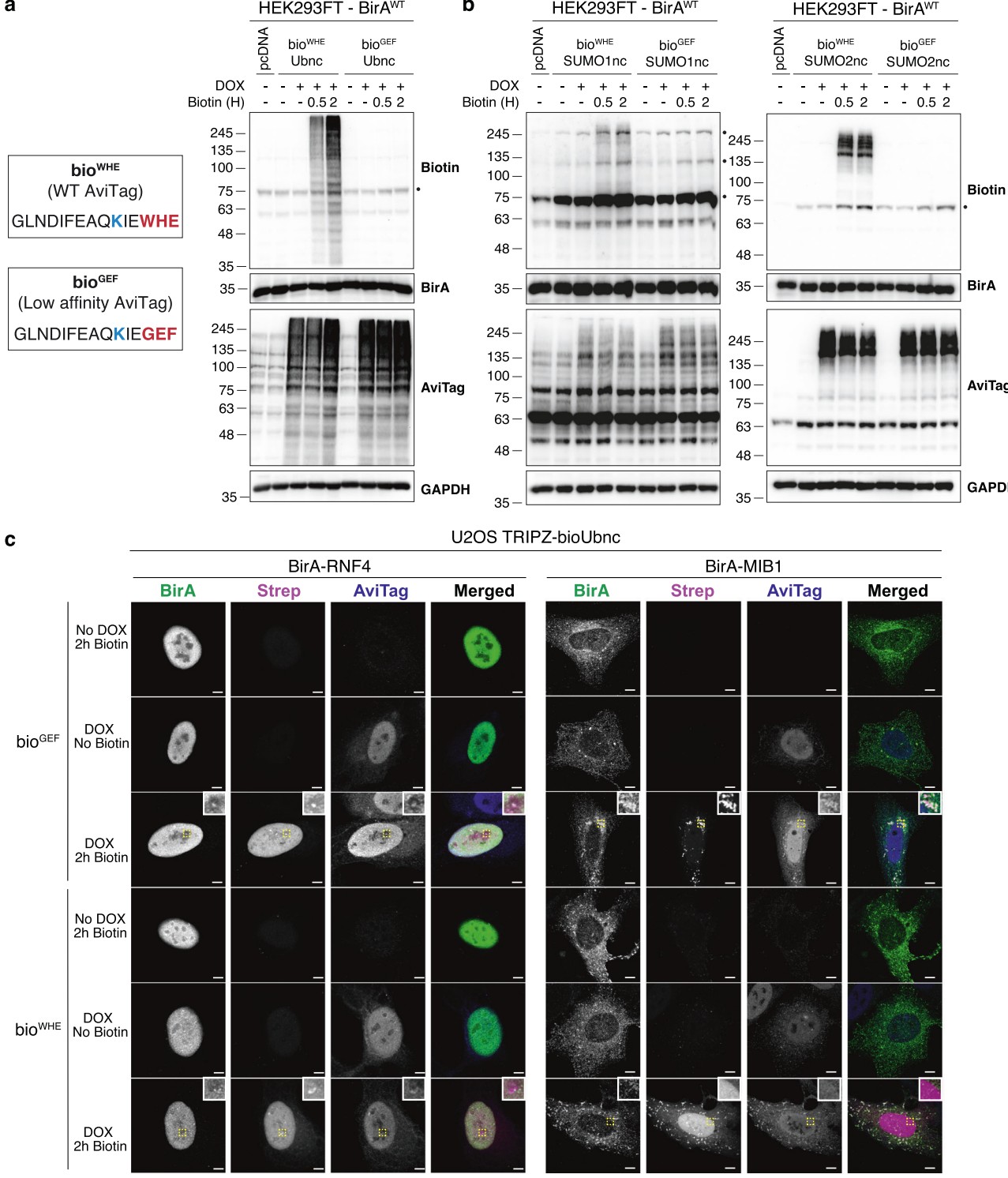

**Fig. 2 | Low affinity bioGEF enables BioE3 studies. a, b** Left, sequence of the WT (WHE) and the low affinity (GEF) AviTags. Biotin-targeted lysine is shown in blue, mutated amino acids in red. Western blot of HEK293FT stable cell lines expressing EFS-BirA, transfected with **a** TRIPZ–bioWHEUbnc or the low affinity version bioGEFUbnc and **b** bioWHESUMO1nc, bioWHESUMO2nc or the low affinity versions bioGEFSUMO1nc, bioGEFSUMO2nc. Cells were preincubated in biotin-free dialyzed FBS-containing media prior to transfections. Doxycycline (DOX) induction was performed at 1 μg/ml for 24 h and biotin supplementation at 50 μM for the indicated time-points. General unspecific biotinylation was observed for bioWHE tagged UbLs, while no biotinylation was observed in the case of the low-affinity bioGEF versions. Dots indicate endogenous carboxylases that are biotinylated constitutively by the cell. Molecular weight markers are shown to the left of the blots in kDa. **c** Confocal microscopy of U2OS stable cell lines expressing TRIPZ-bioWHEUbnc

or bioGEFUbnc transfected with EFS-BirA-RNF4 or EFS-BirA-MIB1. All BioE3 experiments were performed by pre-incubating the cells in dialyzed FBS-containing media prior to transfections, DOX induction at 1 μg/ml for 24 h and biotin supplementation at 50 μM for 2 h, unless otherwise specified. Colocalization of streptavidin and BirA-RNF4/MIB1 signals was observed when using bioGEFUbnc, while general unspecific labeling was detected for bioWHEUbnc. Yellow dotted-line squares show the selected area for digital zooming. Biotinylated material was stained with fluorescent streptavidin (Strep, magenta), BirA (green), and AviTag (blue) with specific antibodies. Black and white panels show the green, magenta, and blue channels alone. Scale bar: 5 μm for RNF4 panels and 8 μm for MIB1 panels. **a–c** Data are representative of 3 independent transfection experiments with similar results. Source data are provided in the Source data file.

fused at the N-terminus of both ligases to minimize any steric effect on the C-terminal RING domains. Confocal microscopy revealed that cells lacking DOX induction showed no expression of bioUbnc (AviTag panels) whereas cells lacking biotin treatment showed no streptavidin labeling (Strep panels). The correct cellular localization was also confirmed for both BirA-RNF4 (nuclear) and BirA-MIB1 (cytoplasmic, centriolar satellites; BirA panels). For both E3 fusions, with DOX induction and biotin labeling, the correct colocalization between the BirA and streptavidin signal was observed when using bio$^{GEF}$Ubnc (Fig. 2c, 3rd row), but non-specific streptavidin signal appeared when using bio$^{WHE}$Ubnc (Fig. 2c, 6th row). Both BirA-E3 fusions could use bio$^{GEF}$Ubnc almost as efficiently as bio$^{GEF}$Ub$^{WT}$ to ubiquitinate and label the substrates, suggesting that the non-cleavable mutant does not impede conjugation for RING-type E3s (Supplementary Fig. 2a). We also tested PEX12, an Ub E3 ligase that specifically localizes to peroxisomes, with PEX12 BioE3 yielding specific BirA and streptavidin colocalization when using bio$^{GEF}$Ubnc, but additional non-specific streptavidin signal in the nucleus with bio$^{WHE}$Ubnc (Supplementary Fig. 2b). Since bio$^{GEF}$ improved specificity, we wanted to test BioE3 using E3 ligases for other UbLs, so we prepared BirA-PIAS1 and BirA-PIAS4 for use with 293FT SUMO2nc cells. As with Ub, we observed that bio$^{GEF}$SUMO2nc showed enhanced specific labeling of PIAS1 and PIAS4 substrates (compare WT versus catalytic mutant CA; Supplementary Fig. 3). Taken together, use of the bio$^{GEF}$ tag with controlled biotin labeling provides the desired specificity to enable the BioE3 method for multiple UbL E3 ligases.

## RNF4 BioE3 specifically targets PML

To test BioE3 specificity for identifying substrates, we decided to use RNF4, a well-characterized SUMO Targeted Ubiquitin Ligase (STUbL), that recognizes SUMOylated substrates through SUMO Interacting Motifs (SIMs) to ubiquitinate and target them for proteasomal degradation[21]. We generated three versions of BirA-RNF4: (1) WT, (2) a catalytically inactive version (CA), with a mutant RING domain to impair its interaction with the E2~Ub, and (3) a version with mutated SIMs (ΔSIM) that impairs its interaction with SUMOylated substrates. We performed BioE3 in 293FT bio$^{GEF}$Ubnc cells, comparing RNF4$^{WT}$, with or without proteasome inhibitor, to RNF4$^{CA}$ and RNF4$^{ΔSIM}$ mutants (Fig. 3a). We posited that biotin-labeled substrates seen with RNF4$^{WT}$ compared to the RNF4$^{CA}$ mutant, especially those that significantly accumulated upon proteasomal inhibition, would constitute the ubiquitinated targets of RNF4 (Fig. 3a, biotin blot and biotin signal quantification graph). Those targets were largely dependent on SUMO-SIM interactions, as BirA-RNF4$^{ΔSIM}$ showed biotinylation similar to the background obtained with RNF4$^{CA}$ (Fig. 3a, biotin blot). We also performed RNF4 BioE3 in the U2OS bio$^{GEF}$Ubnc cells and checked subcellular biotinylation by confocal microscopy (Fig. 3b, Strep panels). The nuclear BirA-RNF4$^{WT}$ correctly colocalized with the streptavidin signal, while the CA and ΔSIM versions showed only background levels of biotinylation (Fig. 3b).

A well-known substrate of RNF4 is PML, which undergoes poly-SUMOylation and subsequent ubiquitination by RNF4 upon cellular exposure to arsenic trioxide (ATO), with the modified PML targeted for proteasomal degradation[22,23]. We performed RNF4 BioE3 in 293FT bio$^{GEF}$Ubnc cells, using mutant controls, with ATO treatment (Fig. 3c). As expected, PML was significantly enriched after treating the cells with ATO, compared to RNF4$^{CA}$ or RNF4$^{ΔSIM}$ (Fig. 3c, quantified 3-fold enrichment). We evaluated RNF4 BioE3 labeling of PML by confocal microscopy in U2OS bio$^{GEF}$Ubnc cells. We observed that, in basal conditions, BirA-RNF4$^{WT}$ biotinylates proteins that localize to the nucleoplasm and some nuclear bodies, but not PML (Fig. 3d). Treatment with ATO, MG132, or both induced the formation of larger PML nuclear bodies, BirA-RNF4 recruitment and biotinylation, likely of Ub-modified targets, with specific colocalization (Fig. 3d). These data support that BioE3 is capable to label a specific target of RNF4 and moreover, in response to a chemical stimulus.

## RNF4 BioE3 identifies many SUMO-dependent targets

Since RNF4 BioE3 could identify PML, we performed large-scale experiments in triplicate comparing RNF4$^{WT}$, RNF4$^{CA}$, and RNF4$^{ΔSIM}$, confirmed the samples by western blot (WB, Supplementary Fig. 4) and processed the streptavidin pull-down eluates by LC-MS in order to identify the specific targets of RNF4. 188 proteins were enriched using BioE3 when comparing RNF4$^{WT}$ to its CA version (Fig. 4a, Supplementary Data 1). Among them, many proteins related to the Ub machinery were identified, including E1 activating enzymes (UBA2 and UBA6), E2 conjugating enzymes, E3 ligases, and DUBs, that could represent active Ub carriers that form complexes with RNF4. Some of RNF4 substrates might be components of PML NBs, so we compared our list of RNF4 targets to lists of potential PML NB components identified previously by proximity labeling or YFP-PML pull-down MS[24,25]. In total, 37 of the potential targets of RNF4 associate with PML NBs (Supplementary Data 1).

We also compared BioE3 of RNF4$^{WT}$ and RNF4$^{ΔSIM}$, to estimate the percentage of SUMO-dependent substrates. In this case, BioE3-RNF4$^{WT}$ identified 205 proteins, most of them being also enriched when comparing RNF4$^{WT}$ to RNF4$^{CA}$ (Fig. 4b, Supplementary Data 1). In total, 124 out of the 188 (66%) substrates appear to be SUMO-dependent targets of RNF4, indicating that SUMO-SIM-dependent substrate recognition is the prevalent mode of RNF4 recruitment (Fig. 4b–d). It is worth mentioning that SUMO1 and SUMO2 peptides were highly enriched in both RNF4$^{WT}$/RNF4$^{CA}$ and RNF4$^{WT}$/RNF4$^{ΔSIM}$ BioE3s (Fig. 4a, b). Furthermore, BioE3-RNF4$^{WT}$ eluates were highly enriched in SUMO2/3 modified proteins compared to both RNF4$^{CA}$-BirA and RNF4$^{ΔSIM}$ (Fig. 4c), showing the high specificity of BioE3 to purify SUMO-dependent Ub targets of RNF4. We compared our putative RNF4 targets with a comprehensive dataset of SUMOylated proteins[26], and concluded that 91% were part of the SUMOylome (Fig. 4d, Supplementary Data 1).

RNF4 shows SIM-dependent accumulation at DNA damage sites, which are also loci of SUMO-dependent signaling[27,28]. Two SUMO-dependent targets identified by BioE3, Fanconi Anemia group I protein, FANCI and FANCD2, were shown to be SUMOylated on damaged chromatin and regulated through ubiquitination by RNF4 to allow cell survival after DNA damage[29]. MDC1 also participates in DNA repair and was previously shown to be a SUMO-dependent target of RNF4[30]. Interestingly, MDC1 SUMOylation regulates homologous recombination through TP53BP1, which was also detected as RNF4 target by BioE3. Also linked to DNA repair, PARP1 has previously been identified as an interactor and SUMO-dependent substrate of RNF4[31,32].

In conclusion, these results show that BioE3 is highly specific and sensitive enough to identify E3 substrates, as exemplified by the SUMO-dependent targets of RNF4.

## RNF4 E3 ligase activity regulates essential nuclear and Ub/proteasome-related processes

To assess the functional role of RNF4 Ub E3 ligase activity, we performed STRING network analysis using the 188 potential RNF4 targets. The network showed a major interconnected core-cluster composed of 89% of the identified substrates (Supplementary Fig. 5). Unsupervised MCODE analysis highlighted 5 main derived sub-clusters composed of proteins related to RNA processing, DNA repair, the ubiquitin-proteasome system (UPS), DNA recombination and damage response, replication and translation (Fig. 5a). Furthermore, gene ontology (GO) analysis highlighted processes related to replication, RNA binding, UPS, DNA repair and cell cycle regulation (Fig. 5b, Supplementary Data 2). The DNA replication machinery is particularly regulated by RNF4, as many components of the replication fork and proteins with helicase activity, e.g., Cdc45-MCM-GINS (CMG) and the Mini-Chromosome Maintenance (MCM) complexes, have been identified by BioE3.

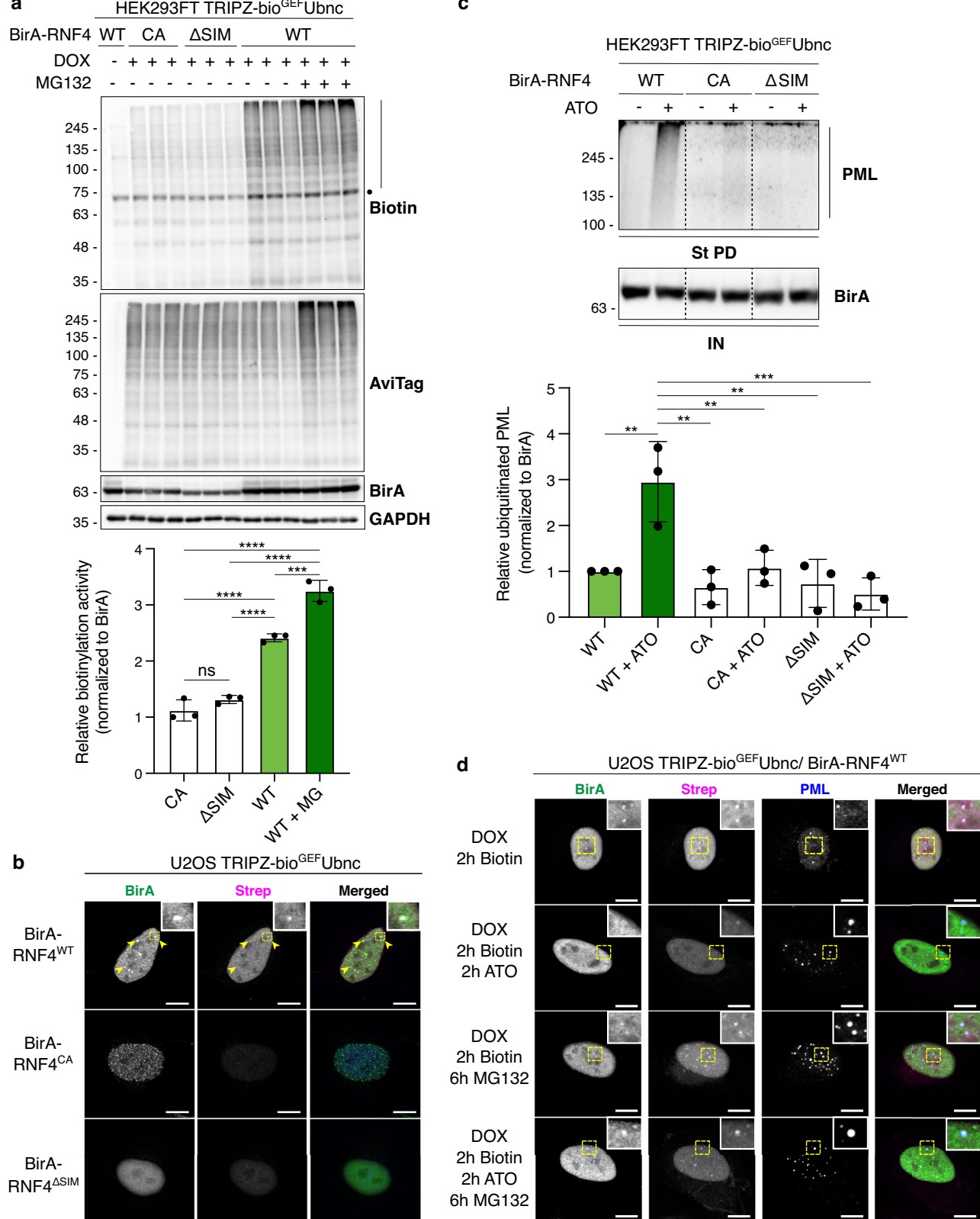

## BioE3 of MIB1 points to regulation of centrosomes and autophagy

To further assess the ability of BioE3 to identify targets of RING type Ub E3 ligases, we applied this strategy to MIB1, an E3 ligase involved in Notch signaling pathway[33,34] and known to localize to centriolar satellites[35,36]. We generated constructs to express BirA-MIB1[WT] or its CA

version, and tested BioE3 in 293FT bio[GEF]Ubnc cells by WB. We observed strong and specific BioE3 activity for MIB1[WT] at 2 and 4 h of labeling compared to its CA counterpart (Fig. 6a). Subcellular localization of BioE3-MIB1 activity was checked in U2OS bio[GEF]Ubnc cells and we observed that biotinylation colocalizes with BirA-MIB1[WT] at centrosomes, as well as in vesicle-like structures (Fig. 6b, Strep panel). The

**Fig. 3 | BioE3 specifically labels substrates of RNF4. a** Western blot of BioE3 experiment in triplicates performed on HEK293FT stable cell line expressing TRIPZ-bio$^{GEF}$Ubnc and transfected with EFS-BirA-RNF4$^{WT}$, BirA-RNF4$^{CA}$, or BirA-RNF4$^{\Delta SIM}$. MG132 was used at 10 μM for 4 h. Specific biotinylation of RNF4 targets, which were accumulated upon MG132 treatment (bars), was observed. Dot indicates endogenously biotinylated carboxylases. Biotin signal was quantified and normalized to expression levels (BirA blot). **b, d** Confocal microscopy of BioE3 experiment performed on U2OS stable cell line expressing TRIPZ-bio$^{GEF}$Ubnc transfected with EFS-BirA-RNF4$^{WT}$, BirA-RNF4$^{CA}$ or BirA-RNF4$^{\Delta SIM}$. Yellow dotted-line squares show the selected colocalization event for digital zooming. Biotinylated material is stained with fluorescent streptavidin (Strep, magenta), and BirA with specific antibody (green). In blue, nuclei are labeled with DAPI (**b**) or PML is labeled with specific antibody (**d**). Black and white panels show the green and magenta channels individually. Scale bar: 10 μm. Colocalization of streptavidin and BirA-RNF4$^{WT}$ signals was observed in the nucleus (yellow arrowheads) (**b**). Indicated samples were also treated with 1 μM ATO for 2 h and 10 μM MG132 for 6 h (**d**). **c** Western blot showing the effect of ATO treatment in PML ubiquitination (black arrowhead and bars) by RNF4. BioE3 experiment was performed on HEK293FT stable cell line expressing TRIPZ-bio$^{GEF}$Ubnc transfected with EFS-BirA-RNF4$^{WT}$, BirA-RNF4$^{CA}$ or BirA-RNF4$^{\Delta SIM}$. Indicated samples were also treated with 1 μM ATO for 2 h. St PD, streptavidin pull-down. PML signal in the pull-down was quantified and normalized to expression levels (BirA blot). **a, c** Bar plots show the mean and standard deviation of $n = 3$. Statistical analyses were performed by one-way ANOVA with Tukey's multiple comparison test: **$p < 0.01$; ***$p < 0.001$; ****$p < 0.0001$. Source data and the exact $p$-values are provided in the Source Data file. Molecular weight markers are shown to the left of the blots in kDa, antibodies used are indicated to the right. Source data and the exact $p$-values are provided in the Source data file. **a–d** Data are representative of 3 independent experiments with similar results. Source data are provided in the Source data file.

CA version has similar localization, but no biotinylation activity was observed (Fig. 6b).

We then performed a large-scale MIB1 BioE3 experiment for analysis by LC-MS. In total, 57 proteins were enriched in bio$^{GEF}$Ubnc MIB1$^{WT}$ BioE3 compared to MIB1$^{CA}$ (Fig. 6c, Supplementary Data 3). Among them, centrosomal-associated proteins such as PCM1, CEP131, USP9X, and CYLD were identified, as well as CP110 with lower confidence[37], consistent with the fact that MIB1 localizes to centriolar satellites, pericentriolar material and centrosomes. We compared BioE3 MIB1 substrates to a published MIB1 proximity labeling dataset[36], and found that 19 proteins (33%) are high confidence MIB1 Ub substrates, among which the previously mentioned centrosomal proteins are present (Supplementary Fig. 6a, Supplementary Data 3). Importantly, similar and comparable expression levels of BirA-MIB1$^{WT}$/MIB1$^{CA}$ were confirmed (Fig. 6d, BirA blot). We further confirmed PCM1, USP9X and CEP131 as MIB1 Ub substrates by WB (Fig. 6d).

We performed STRING network analysis on the 57 identified significant substrates of MIB1. 67% of the proteins formed an interconnected core-cluster, from which the major sub-clusters were related to endocytosis and autophagy, containing TAB1, NBR1, OPTN, HGS, SQSTM1, STAM2, and CALCOCO2 (Supplementary Fig. 6b; Fig. 6c). GO analysis highlighted Ub and UPS related processes, due to presence of Ub E2 conjugase UBE2S and DUBs (USP24, CYLD, UCHL1 and UCHL3), as well as hits related to endosomal and vesicular trafficking, autophagy, centrosomes and midbody (Fig. 6e, Supplementary Data 4). Thus, BioE3 enabled the identification of MIB1 substrates and pathways in which its E3 ligase activity is implicated.

## Applying BioE3 to organelle-specific and uncharacterized E3 ligases

To test BioE3 specificity further, we selected an organelle-specific E3 ligase, MARCH5, a RING-type E3 that resides primarily in the mitochondrial outer membrane and has roles in regulating mitochondrial morphology[38]. As before, we generated fusions of the wild-type E3 or its CA version to the BirA enzyme (BirA-MARCH5$^{WT}$ and BirA-MARCH5$^{CA}$). BirA was fused at the N-terminus of MARCH5, and even if the RING domain is close to N-terminus, previous N-terminal tagging using FLAG has been reported[39]. We tested the system in 293FT bio$^{GEF}$Ubnc cells by WB and observed specific biotinylation of proteins after 2 and 4 h of biotin treatment by BirA-MARCH5$^{WT}$ in comparison with its CA version (Fig. 7a). Furthermore, by confocal microscopy, BirA-MARCH5$^{WT}$ colocalized with biotinylated proteins at mitochondria in U2OS bio$^{GEF}$Ubnc cells (Fig. 7b). The CA counterpart also displays a mitochondrial localization, but the biotinylation levels were dramatically reduced.

We also selected a less characterized RING-type E3, RNF214, to explore the discovery potential of BioE3. BirA was fused at the N-terminus of RNF214 to minimize any steric effect on the C-terminal RING domain. Using 293FT bio$^{GEF}$Ubnc cells, RNF214 BioE3 showed specific biotinylation activity of BirA-RNF214$^{WT}$ in comparison with the CA counterpart by WB (Fig. 7c). RNF214 BioE3 in U2OS bio$^{GEF}$Ubnc cells was analyzed by confocal microscopy and we observed that BirA-RNF214 fusions localize to the cytoplasm, with additional centrosomal enrichment. Biotinylation activity was only observed with BirA-RNF214$^{WT}$, colocalizing with the BirA signal (Fig. 7d).

Considering this pilot data, we posited that BioE3 may specifically label targets of MARCH5 or RNF214. Thus, we performed large-scale BioE3 experiments for analysis by LC-MS. We expected that MARCH5 and RNF214 targets differ significantly based on their different subcellular localization, so we compared the two E3 ligases with each other. We identified 31 putative targets of MARCH5 (Fig. 7e, Supplementary Data 5). Among them, endogenously biotinylated mitochondrial carboxylases (PC, ACACA, PCCA, and MCCC1) were removed from the analysis due to uncertainty of being targets, leading to a reduced list of 27 hits. We found confirmed targets of this E3 ligase, like MFN2[40] and MCL1[41], albeit with lower confidence. Five out of the 27 targets were annotated in Mitocarta 3.0[42] as mitochondrial proteins and 18 (67%) were part of the mitochondrial proximal interactome[43] (Supplementary Fig. 7a, Supplementary Data 5). GO analysis highlighted the mitochondrial outer membrane and the endoplasmic reticulum membrane (Supplementary Fig. 7b, Supplementary Data 6). Furthermore, we confirmed ARFGAP1, a protein associated with the Golgi apparatus[44], but also found by mitochondrial proximity proteomics[43], as a novel Ub target of MARCH5 by WB (Fig. 7f). Importantly, similar and comparable expression levels of BirA-MARCH5$^{WT}$/MARCH5$^{CA}$ and BirA-RNF214$^{WT}$/RNF214$^{CA}$ were also validated (Fig. 7f, BirA blots). This data shows the utility of BioE3 for identifying E3 substrates with possible roles in organelle crosstalk.

RNF214 BioE3 identified 109 target proteins (Fig. 7e, Supplementary Data 5), and to determine the processes in which RNF214 participates we performed a STRING network analysis. 81% of the proteins formed an interconnected core-cluster, from which 4 main subclusters were derived by unsupervised MCODE analysis (Supplementary Fig. 8). Processes related to translation and metabolism were highlighted. Furthermore, GO analysis showed that RNF214 plays a key role in processes related to cell adhesion, microtubules, translation, and ubiquitination (Supplementary Fig. 9a, Supplementary Data 6). We compared those targets to a previously published proximity labeling of RNF214[45] and defined 60 high confident RNF214 Ub substrates (Supplementary Fig. 9b, Supplementary Data 5). We validated by WB several hits of the RNF214 BioE3 as Ub targets of RNF214 (Fig. 7f), supporting the implication of the E3 ligase in the aforementioned processes: ROCK1, a kinase that regulates actin cytoskeleton[46] and GIGYF2, which has a role in translation[47]. We also validated CLINT1, a protein involved in intracellular trafficking[48], as a ubiquitin target of RNF214, even if CLINT1 was below the confidence threshold. These results support the notion that BioE3 can identify novel substrates for poorly characterized E3 ligases.

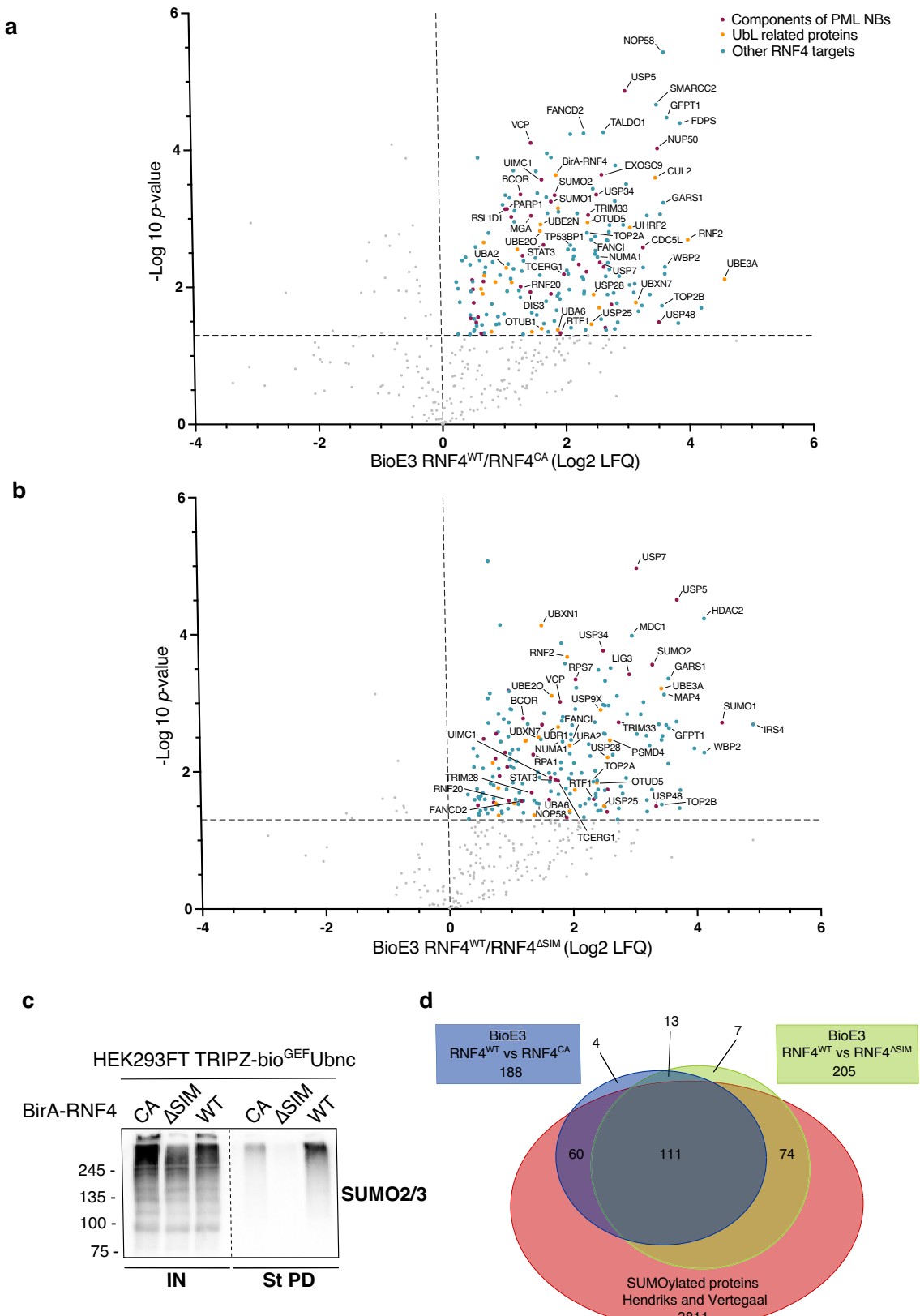

## Engineering BioE3 to study HECT E3 ligases

The successful application of BioE3 to identify substrates of RING-type E3 ligases led us to question whether it could also work for a distinct class, the HECT-type E3 ligases. HECT ligases employ an extra trans-thiolation step in which Ub is passed from E2 to the E3 itself, before transferring to substrates. Some HECT ligases (e.g., NEDD4 subgroup) are also autoinhibited through intramolecular contacts and need activation signals[49]. Using NEDD4 as a candidate, we either removed its N-terminal autoinhibitory C2 domain (NEDD4$^{\Delta C2}$) or we mutated selected residues in the C2 and the HECT domain (generating the hyperactive NEDD4;$^{3M}$ I36A, L37A and Y604A)[50] (see Supplementary Note 2 and Supplementary Fig. 10a–c). The activated NEDD4$^{3M}$ was

**Fig. 4 | BioE3 identifies SUMO-dependent Ub targets of RNF4. a, b** Volcano plots of LC-MS analysis comparing streptavidin pull-downs of BioE3 experiments performed on HEK293FT stable cell line expressing TRIPZ-bio$^{GEF}$Ubnc transfected with EFS-BirA-RNF4$^{WT}$, BirA-RNF4$^{CA}$ or BirA-RNF4$^{\Delta SIM}$, with 3 biological replicates per condition performed ($n = 3$). Proteins significantly enriched (Log 2 RNF4$^{WT}$/ RNF4$^{CA}$ (**a**) or RNF4$^{\Delta SIM}$ (**b**) > 0 and $p$-value < 0.05) were considered as RNF4 targets. Statistical analyses were done using two-sided Student's $t$-test. Data are provided as Supplementary Data 1. **c** Western blot of SUMOylated RNF4 targets from samples described in (**a, b**). IN: input; St PD: streptavidin pull-down. Molecular weight markers are shown to the left of the blots in kDa. **d** Venn diagram showing the SUMO-dependent targets of RNF4 (comparison of the BioE3 RNF4$^{WT}$/RNF4$^{CA}$ targets in (**a**) *versus* the BioE3 RNF4$^{WT}$/RNF4$^{\Delta SIM}$ targets in (**b**)) and the SUMOylated targets (SUMOylome from Hendriks and Vertegaal[26]). Comparison data are provided as Supplementary Data 1. Source data are provided in the Source data file.

confirmed to be properly folded by gel filtration experiments (Supplementary Fig. 10d, e). BirA was fused at the N-terminus of NEDD4, as the location of any tag in the C-terminal part abrogates the catalytic activity of the enzyme[51]. Although BirA-fusions of both NEDD4 mutants showed potential BioE3 activity, the activity-to-background ratio was still low, which prompted us to seek for further improvements. A previous report suggested that Ub mutant L73A is poorly transferred from the E2 to E3 enzyme[52]. We wondered if L73P mutation in Ubnc used in all preceding experiments could be affecting the transthiolation step in the case of NEDD4 and, therefore reducing BioE3 efficiency. We confirmed this hypothesis using an in vitro transthiolation reaction, and observed a clear delay in the discharge of E2-Ub and the formation of HECT-Ub adduct when the Ubnc is used (Fig. 8a). Equal usage of WT and L73P Ub was confirmed by Coomassie staining since recognition by the anti-Ub antibody was partially impaired by the L73P mutation itself (Fig. 8b). Due to inefficient Ub loading of Ubnc (L73P), the enzymatic activity of both NEDD4$^{WT}$ and NEDD4$^{3M}$ hyperactive mutant is severely affected, as shown by in vitro autoubiquitination reaction (Fig. 8c). Of note, we tried to induce NEDD4$^{WT}$ activity using ionomycin and CaCl$_2$ treatment, but only observed weak biotinylation, perhaps due to the Ubnc (L73P) issue (see Supplementary Note 2 and Supplementary Fig. 10b). Therefore, the use of Ub$^{WT}$ could improve the efficiency of NEDD4 BioE3.

Next, we tested NEDD4 BioE3 in 293FT bio$^{GEF}$Ub$^{WT}$ cells, using different versions of BirA-NEDD4 (WT, CA, ΔC2, ΔC2/CA). As expected, autoinhibited NEDD4$^{WT}$ BioE3 appeared similar to NEDD4$^{CA}$, with some auto-ubiquitinated NEDD4 detectable (Fig. 8d). In contrast, NEDD4$^{\Delta C2}$ BioE3 activity was greatly enhanced compared to NEDD4$^{\Delta C2,CA}$, probably attributable to autoubiquitination of NEDD4$^{\Delta C2}$, while the background biotinylation levels using NEDD4$^{\Delta C2,CA}$ were comparable to NEDD4$^{CA}$ (Fig. 8d). Similar results were obtained when performing BioE3 NEDD4$^{3M}$ versus NEDD4$^{3M,CA}$, with improved BioE3 activity-to-background signal ratios (Fig. 8e). In this case, cells were also treated with the DUB inhibitor PR619 (to potentially reduce recycling of bioUb), but no significant differences were observed in patterns of biotinylated bands. We then checked the subcellular localization of BioE3 NEDD4 by confocal microscopy using U2OS bio$^{GEF}$Ub$^{WT}$ cells with WT, ΔC2, and 3M versions of NEDD4 (Fig. 8f). Compared to autoinhibited WT and partially activated ΔC2, the fully-activated version NEDD4$^{3M}$ yielded strong streptavidin signal that correlated with BirA and accumulated in cytoplasmic structures that might correspond to trafficking vesicles (Fig. 8f). Collectively, these results show that BioE3 NEDD4 efficiency is improved when using activating mutations and bio$^{GEF}$Ub$^{WT}$, which may permit target identification for HECT E3s, at least of the NEDD4 subclass.

### BioE3 identifies NEDD4 substrates
We performed a large-scale NEDD4 BioE3, comparing the activated 3 M version to its corresponding transthiolation mutant 3 M/CA in 293FT bio$^{GEF}$Ub$^{WT}$ cells. We identified 59 proteins as potential Ub substrates of NEDD4 (Fig. 8g, Supplementary Data 7). In line with known biological function of NEDD4, many of them were related to vesicular transport and endocytosis such as AMOTL2, PDCD6IP/ALIX, SCAMP3, DUSP1, VPS33A, CALR or the GTPases RAB1A, RAB1B and RAB7A (Fig. 8g), components that were enriched after GO analysis (Supplementary Fig. 10d; Supplementary Data 8). A well-known NEDD4 substrate EPS15 was also identified, albeit with lower confidence, and also the

previously described NEDD4 substrate WBP2[53]. Importantly, some hits, such as PDCD6IP[54] and SCAMP3[55] had been described as NEDD4-interacting proteins. NEDD4 contributes to formation of K63-linked ubiquitin chains[56], and with NEDD4 BioE3, we identified the substrate ABRAXAS2, a subunit of a K63 deubiquitinase complex (BRCA1-A)[57], which suggests a potential feedback regulation. We detected multiple components of the TRiC molecular chaperone complex (CCT8, TCP1, CCT6A, CCT3, and CCT4), that was also enriched as GO term (Supplementary Fig. 10f; Supplementary Data 8) and, in fact, CCT4 was recently implicated as a crucial vesicular trafficking regulator[58]. We validated CCT8, as well as TP53BP2, as NEDD4 Ub substrates by WB (Fig. 8h). Importantly, similar and comparable expression levels of BirA-NEDD4$^{3M}$/NEDD4$^{3MCA}$ were validated, with a small increase in stability for the transthiolation mutant NEDD4$^{3MCA}$ (Fig. 8h, BirA blot). Unexpectedly, many components of the Ub machinery, including the E1 activating enzyme UBA1, multiple E2s and distinct HECT E3 ligases (UBE3A, BIRC6, TRIP12, HERC4) were enriched when performing BioE3 with the transthiolation mutant NEDD4$^{3M,CA}$ (Fig. 8g). We speculate that NEDD4$^{3M,CA}$ can still form the complexes required for ubiquitination and, since the transthiolation step is impaired, bio$^{GEF}$Ubs on the engaged client E2s become biotinylated, leading to recycling.

In sum, these data show that BioE3 can be adapted and applied to HECT E3 ligases. NEDD4 BioE3 successfully identified specific Ub targets of the ligase, supporting its fundamental roles in the regulation of proteins related to endocytosis and vesicular trafficking.

### Orthogonal validations of BioE3 targets
To further support that targets were indeed ubiquitylated by the different E3 ligases that we studied, we used orthogonal approaches for validation. We performed 6xHIS-Ub pulldown experiments, comparing wild-type and catalytic mutant versions of the different BirA-E3s used in this study to confirm ubiquitination status of selected hits (e.g., SUMO-conjugates:RNF4; USP9X/CEP131:MIB1; ARFGAP1:MARCH5; ROCK1:RNF214; CCT8/TP53BP2:NEDD4-3M; Supplementary Fig. 11a). In addition, we generated MIB1 knockout (MIB1-KO) cells, or depleted MIB1 using RNAi, and then enriched the endogenous ubiquitome using TUBEs, with and without proteasome inhibition by bortezomib[59]. Using this approach, we could validate CEP131 and USP9X as Ub targets of MIB1, as the endogenously ubiquitinated fraction of those hits were reduced when removing or reducing MIB1 (Supplementary Fig. 11b, c). In addition, we validated WBP2 as NEDD4 target by in vitro ubiquitination assay (Supplementary Fig. 11d, e). Taken together, these orthogonal assays and validations support that candidate substrates identified by BioE3 are promising leads for future biological studies.

### Discussion
Understanding substrate recognition by particular E3 ligases, as well as identification of their specific targets, are relevant areas of research in the Ub field. To pursue the latter, the expression of an E3 of interest can be manipulated in cells, either reduced by RNA interference/CRISPR or increased by exogenous expression, with subsequent LC-MS evaluation of the total ubiquitome, via enrichment using ubiquitin-specific antibodies (including Ub remnant antibodies i.e., di-Gly/Ubi-SITE), tagged-ubiquitin or TUBEs[59]. However, matching E3s to targets using these approaches can be problematic, failing to distinguish between primary and secondary effects, missing low-level modified

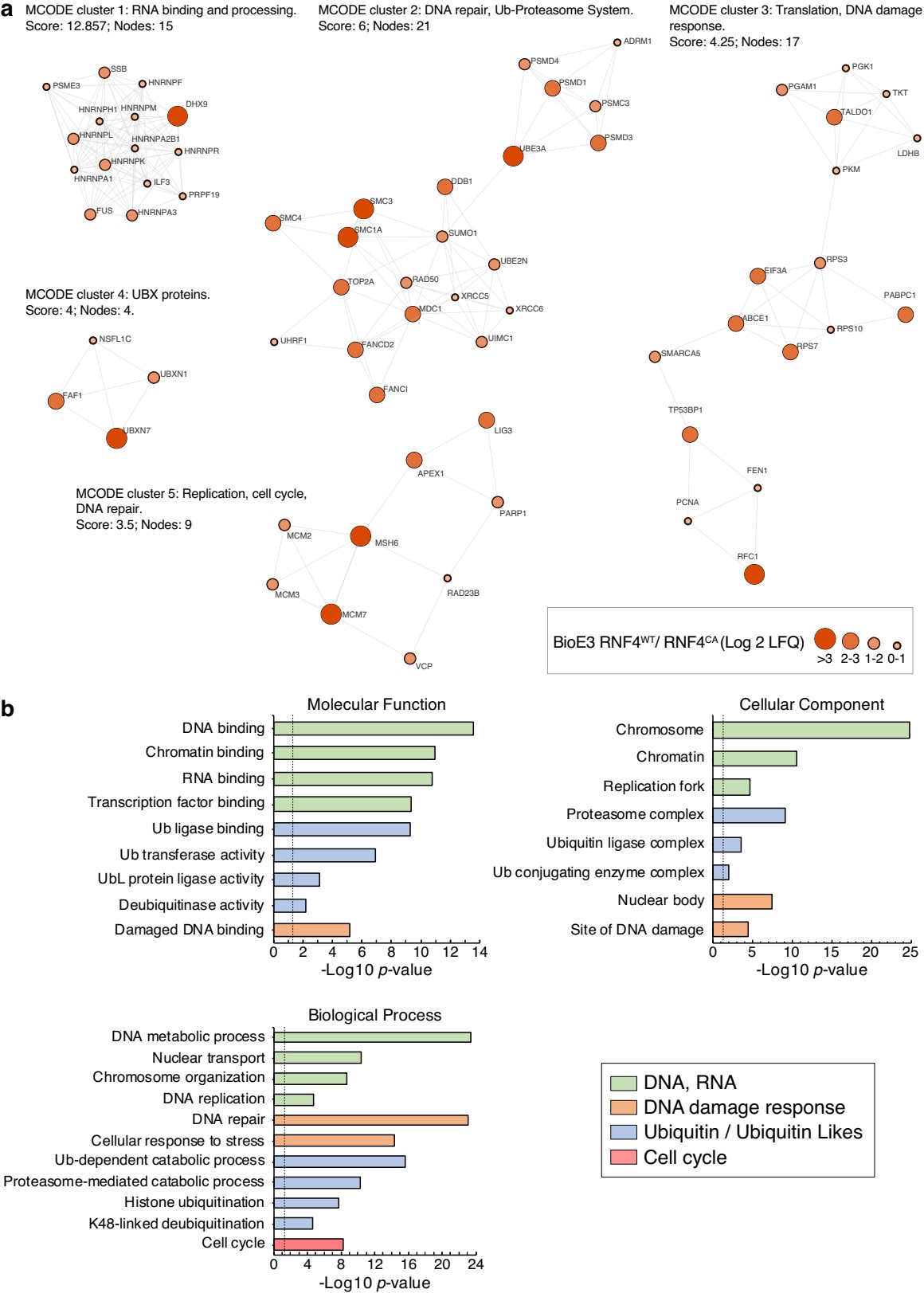

Fig. 5 | RNF4 Ub targets participate in essential nuclear and UPS-related processes. a STRING network analysis of the RNF4 targets defined in Fig. 4a (BioE3 RNF4$^{WT}$/RNF4$^{CA}$). Highly interconnected sub-clusters were derived from the core-cluster in Supplementary Fig. 5 using MCODE. Color, transparency, and size of the nodes were discretely mapped to the Log2 enrichment value as described. b Gene ontology analysis of the RNF4 targets defined in Fig. 4a (BioE3 RNF4$^{WT}$/ RNF4$^{CA}$).

Statistical enrichment analysis was performed using Fisher's one-tailed test with g:SCS correction for multiple comparisons. Depicted biological processes, molecular functions, and cellular components were significantly enriched. Dotted line represents the threshold of the $p$-value (0.05). Data are provided as Supplementary Data 2.

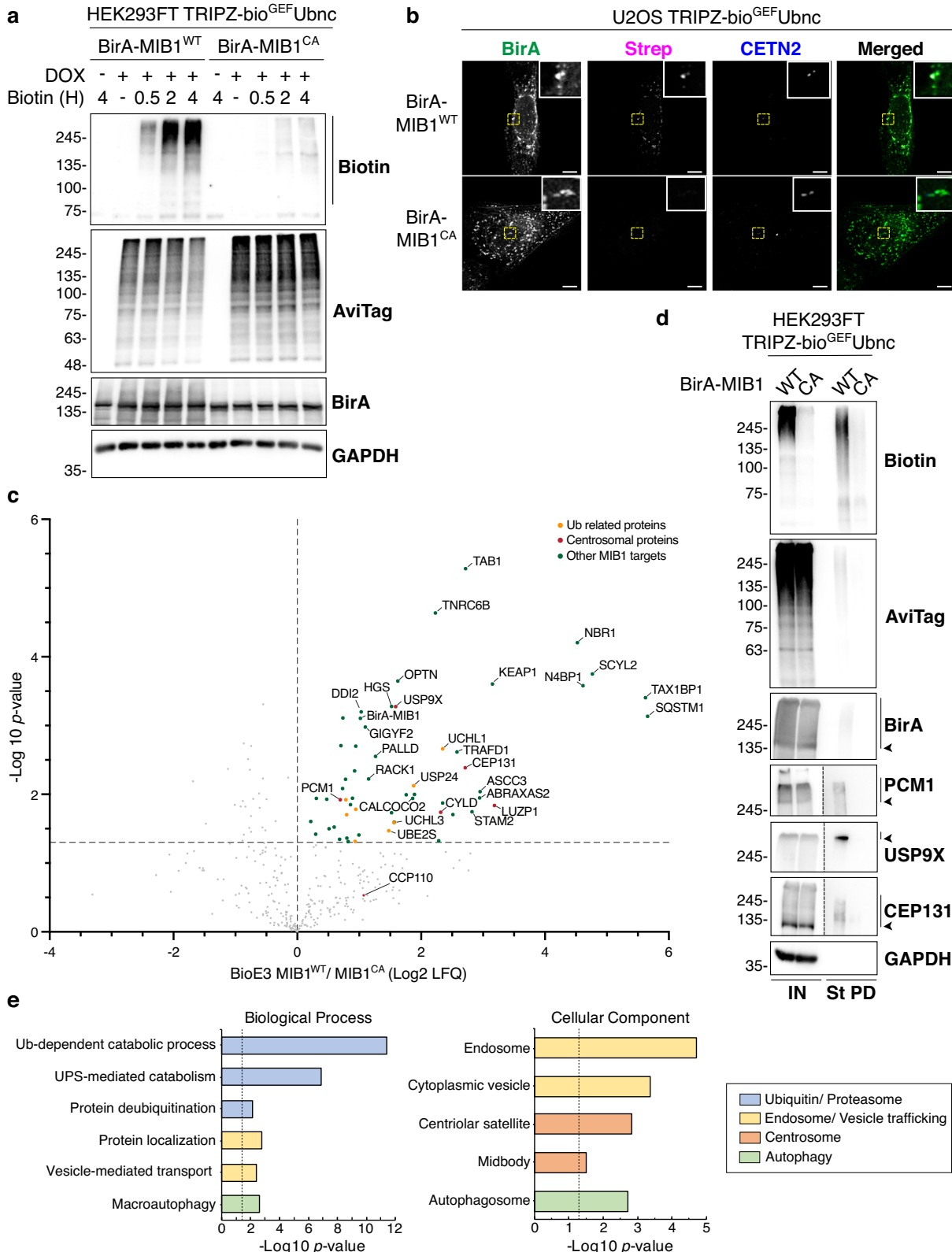

substrates, and capturing non-covalent Ub interactors as false positives. When applied to E3s, BioID-based proximity proteomics[60] can also serve to identify potential substrates, but will equally identify non-covalent interactors or nearby components of protein complexes. Fusions between E3 ligases and UbLs (UBAIT, TULIP, SATT)[11–13] yield promising candidate substrates, but can be limiting due to E3 ligase size and unexpected effects of creating non-physiological E3-UbL-

substrate fusions in cells. Fusions between E3s and ubiquitin-binding domains[14] also show promise, but may have bias for polyubiquitinated substrates and yield false positives like polyUb substrates arising from other proximal E3s.

Complementary to these mentioned approaches, BioE3 is a powerful method to label and identify specific substrates of Ub E3 ligases in vivo. By harnessing BirA-E3 fusions for proximity-dependent

**Fig. 6 | BioE3 identifies targets of MIB1. a** Western blot of BioE3 experiment performed on HEK293FT stable cell line expressing TRIPZ-bio$^{GEF}$Ubnc and transfected with EFS-BirA-MIB1$^{WT}$ or BirA-MIB1$^{CA}$. Specific biotinylation of MIB1 targets was observed at different biotin timings (bars). Molecular weight markers are shown to the left of the blots in kDa. **b** Confocal microscopy of BioE3 experiment performed on U2OS stable cell line expressing TRIPZ-bio$^{GEF}$Ubnc transfected with EFS-BirA-MIB1$^{WT}$ or BirA-MIB1$^{CA}$. Colocalization of streptavidin (Strep, magenta), BirA-MIB1 (green), and Centrin-2 (CETN2, blue) was observed at the centrosomes and selected for digital zooming (yellow dotted-line squares). Black and white panels show the green, magenta, and blue channels individually. Scale bar: 8 µm. **c** Volcano plot of LC-MS analysis comparing streptavidin pull-downs of BioE3 experiments performed on HEK293FT stable cell line expressing TRIPZ-bio$^{GEF}$Ubnc and transfected with EFS-BirA-MIB1$^{WT}$ or BirA-MIB1$^{CA}$ ($n$ = 3 biological replicates per condition). Proteins significantly enriched (Log2 MIB1$^{WT}$/MIB1$^{CA}$ > 0 and $p$-value < 0.05) were considered as MIB1 targets. Statistical analyses were performed by two-sided Student's $t$-test. Data are provided as Supplementary Data 3. **d** Western blot validations of centrosomal MIB1 targets identified in (**c**): PCM1, USP9X, and CEP131. Arrowheads and bars point to unmodified and Ub modified proteins, respectively. IN: input; St PD: streptavidin pull-down. Molecular weight markers are shown to the left of the blots in kDa. **e** Gene ontology analysis of the MIB1 targets defined in (**c**). Statistical enrichment analysis was performed using Fisher's one-tailed test with g:SCS correction for multiple comparisons. Depicted biological processes and cellular components were significantly enriched. Dotted line represents the threshold of the $p$-value (0.05). Data are provided as Supplementary Data 4. **a**–**d** Data are representative of 3 independent transfection experiments with similar results. Source data are provided in the Source data file.

site-specific labeling of bioUb, with attention to recycling, expression levels and biotin availability, BioE3 proves to be highly specific for tagging, purifying, and identifying direct targets for particular E3s. The bio$^{GEF}$-UbLs are only slightly larger than endogenous UbLs, reducing steric effects, and BirA-E3 fusions do not remain engaged to substrates. While we only tested N-terminal BirA fusions to E3 ligases in this study, we expect that activity and substrates identified could vary depending on linker length and positioning of the BirA-tagging relative to the E3 (N-terminal, C-terminal, or even internal). This should be decided depending on the information available for a particular E3 of interest. Exogenous expression of BirA-E3s is used, although lower levels could be achieved using selection of stable lines or inducible expression, with corresponding scale-up in cell numbers to achieve sufficient material for mass spectrometry. Since bio$^{GEF}$ modifies the Ub N-terminus, the method might work less efficiently for linear chain-specific E3s; bio$^{GEF}$Ub could incorporate as single, chain-terminating modifier. BioE3 should enable identification of monoubiquitinated and other classes of polyubiquitinated substrates, as bioUb has been described to generate the different types of chains[10]. We demonstrate here that BioE3 can be applied to different types of ligases (RING, HECT), soluble or membrane-associated, or in different subcellular compartments (nucleus/nuclear bodies, mitochondria, centrosomes). BioE3 can be adapted to most cell lines, and allows processing of lysates for WB or LC/MS, as well as microscopic analysis. This method may be used to follow stimuli-dependent activation or substrate recognition of E3s (e.g., ATO and RNF4, ionomycin, and NEDD4). Importantly, BioE3 detects direct bona fide targets of E3s, in contrast to indirect targets or non-covalent interactors of the E3s.

We showed the applicability of BioE3 to identify Ub targets of RING-type E3 ligases, the largest family of Ub E3 ligases. Concordant with the literature, we found that RNF4 targets are implicated in essential nuclear processes like DNA damage response[61–64], chromosome organization[65], and replication[66,67], among others. In addition, RNF4 targets coincide with PML NBs, in line with the observation that inhibiting ubiquitination causes accumulation of SUMOylated proteins in PML NBs[68]. BioE3 was able to follow the targeting of PML by RNF4 in response to ATO-induced SUMOylation, suggesting that the method is able to monitor changes in E3 targets during chemical treatments, a promising feature for emerging strategies in drug-induced targeted protein degradation (TPD).

MIB1 E3 ligase activity has been linked primarily to the regulation of Notch signaling[69,70] and proximity proteomics has supported roles in endosomal and vesicular trafficking, Ub modifications and cell adhesion[36]. Among the top MIB1 BioE3 hits, we identified several Selective Autophagic Receptors (SARs)[71], particularly from the p62/SQSTM1-like receptor (SLR) class, that culminate in selective-autophagy[72]. Specifically, MIB1 BioE3 identified 5 of 6 known SLRs: NBR1, SQSTM1, OPTN, TAX1BP1, and CALCOCO2. These proteins have well-characterized ubiquitin-binding motifs, with some evidence of direct ubiquitination; MIB1 may ubiquitinate them directly. Therefore, our data add further support for MIB1 as a regulator of autophagy[73].

In concordance with its role in centriolar satellites[74,75], MIB1 BioE3 identified centrosomal and pericentriolar proteins as high confidence targets. Interestingly, MIB1 ubiquitination of PCM1 was shown to be counteracted by USP9X and CYLD, to maintain centriolar satellite integrity[76–78]. In fact, CYLD was shown to directly deubiquitinate auto-ubiquitinated MIB1, inducing its inactivation[78]. Our results support that MIB1 ubiquitination of USP9X and CYLD may contribute in a feedback loop to regulate aspects of centrosomal proteostasis.

To address whether BioE3 could identify substrates of a membrane-localized organelle-specific E3, we chose MARCH5, known to regulate mitochondrial and endoplasmic reticulum contacts through K63 ubiquitination of MFN2[40,79]. While MFN2 was identified with low confidence, we validated the high confidence hit ARFGAP1, a GTPase-activating protein that promotes uncoating of Golgi-derived COPI-vesicles[80]. Ubiquitination as a mechanism for regulating organelle contacts is still largely unexplored. Along with MARCH5 BioE3, we decided to query RNF214, a little-studied E3 ligase (of which there are many), to explore the discovery potential of the method. A systematic BioID study[45] identified proximal partners of RNF214 linked to mRNA biology, translation, microtubules, and actin cytoskeleton, and this was further supported by our BioE3 results. This highlights that BioE3 can discriminate between close interactors and potential direct targets of E3s, focusing the attention on a shorter, more specific list of candidate substrates.

We further showed BioE3 applicability to identify targets of HECT E3s. Additional challenges are present when trying to identify substrates of this type of E3s, because HECTs are often big proteins with signal-dependent activity, with a basal autoinhibited, inactivated state. In the case of NEDD4, we bypassed signals and inhibition by using mutated "active" variants NEDD4$^{\Delta C2}$ and NEDD4$^{3M50,81}$. Together with the use of Ub$^{WT}$ to allow efficient transthiolation, the active mutants showed enhanced BioE3 activity. We believe that the versatile BioE3 method could be used to evaluate the influence of activating/inhibiting mutations, growth factors or other cytokines, or drugs on ligase activity for specific E3s, for monitoring by UbL modification by WB, mass spectrometry, or microscopy.

In summary, we show here that the BioE3 strategy efficiently identifies specific targets of E3 ligases, and could unlock new biology if applied to more of the 600 known E3 ligases, most of which have unknown targets. This is particularly urgent considering the growing relevance of the TPD and its potential application in biomedicine. TPD has significantly evolved in the recent years, with molecular glues approved for the treatment of leukemias and some PROTeolysis-TArgeting Chimeras (PROTACs) to degrade disease-causing proteins undergoing clinical trials, while only a small number of E3 ligases are being employed[82]. BioE3 could assist in characterizing new E3s for use in TPD, identifying on-target and off-target substrates when using TPD strategies, and defining the substrate-recognition properties of E3s through mutant studies, pushing forward TPD innovation by increasing our knowledge of the E3 ligase-substrate network. BioE3 could further be applied to multi-protein complex E3s (e.g., APC/C, SCF,

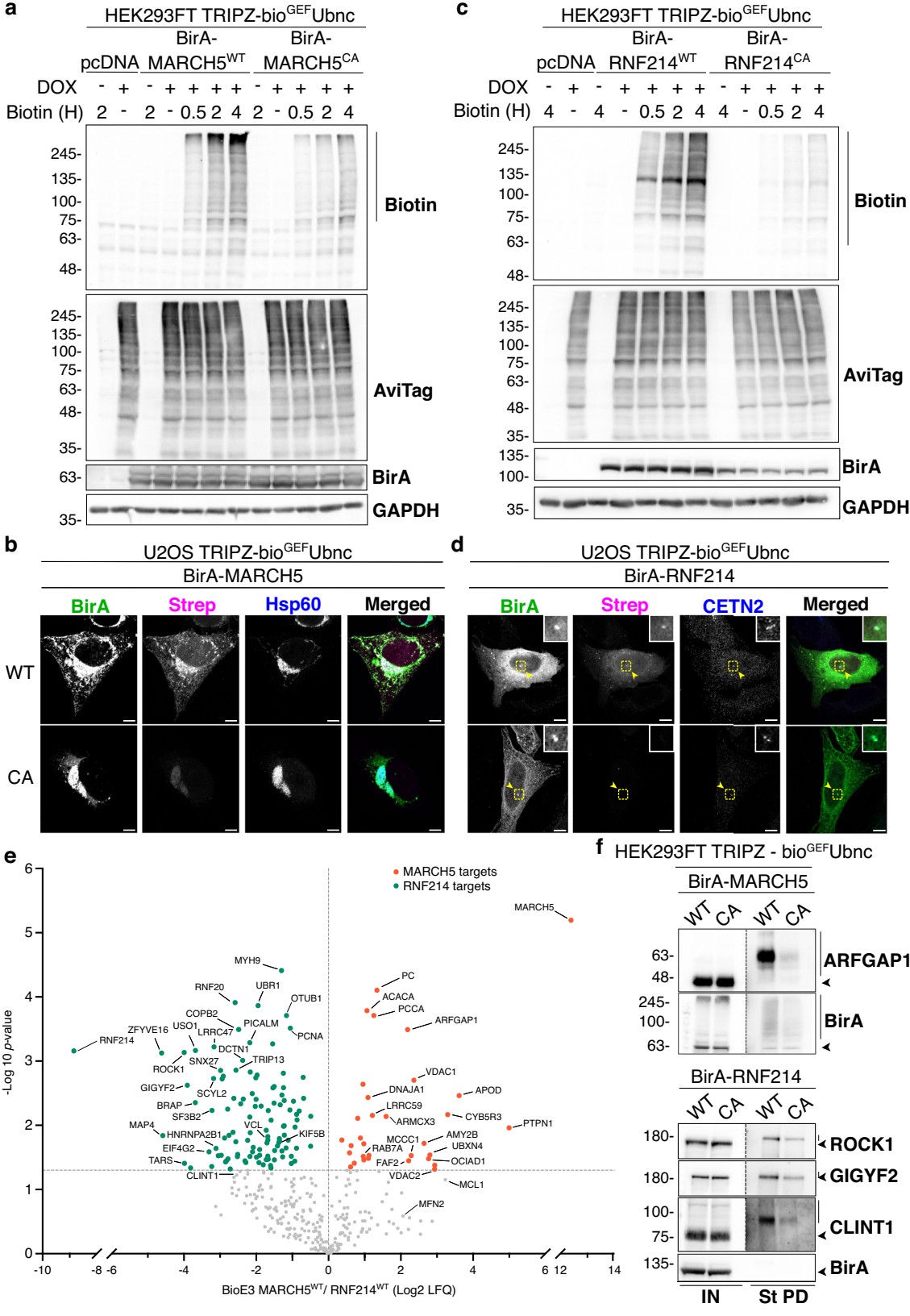

other Cullin RING ligases) to explore substrate specificity, and combining it with subsequent ubiquitination site purification approaches (di-Gly IP, UbiSITE IP), could lead to the identification of Ub sites on substrates carried by specific E3s. With proper bioinformatics analysis, BioE3 could also be used to define consensus sequences and motifs that a particular E3 uses to bind and ubiquitinate the substrate.

## Methods

### Cell culture

U2OS (ATCC HTB-96) and HEK293FT (or 293FT; Invitrogen R7007) were cultured at 37 °C and 5% $CO_2$ in Dulbecco's modified Eagle medium (DMEM) supplemented with 10% fetal bovine serum (FBS, Gibco) and 1% penicillin/streptomycin (Gibco). In general, 293FT cells

**Fig. 7 | BioE3 identifies Ub targets of MARCH5 and RNF214. a**, **c** Western blot of BioE3 experiment performed on HEK293FT stable cell line expressing TRIPZ-bio^GEF^Ubnc and transfected with **a** EFS-BirA-MARCH5^WT^ or BirA-MARCH5^CA^ and **c** EFS-BirA-RNF214^WT^ or BirA-RNF214^CA^. Specific biotinylation of MARCH5 and RNF214 targets was observed at different biotin timings (bars). Molecular weight markers are shown to the left of the blots in kDa. **b**, **d** Confocal microscopy of BioE3 experiment performed in U2OS stable cell line expressing TRIPZ-bio^GEF^Ubnc and transfected with EFS-BirA-MARCH5^WT^ or BirA-MARCH5^CA^ (**b**) and EFS-BirA-RNF214^WT^ or BirA-RNF214^CA^ (**d**). Colocalization of streptavidin (Strep, magenta) and BirA (BirA antibody, green) signals was observed at mitochondria (Hsp60, blue) (**b**) or at the centrosome (Centrin-2, CETN-2, blue) (**d**). Black and white panels show the green, magenta, and blue channels individually. Scale bar: 8 μm. Yellow dotted-line squares show the selected colocalization event for digital zooming. **e** Volcano plot of LC-MS analysis comparing streptavidin pull-downs of BioE3 experiments performed on HEK293FT stable cell line expressing TRIPZ-bio^GEF^Ubnc transfected with EFS-BirA-MARCH5^WT^ and BirA-RNF214^WT^ (n = 3 biological replicates). Proteins significantly enriched (p-value < 0.05) were considered as targets. Statistical analyses were performed by two-sided Student's t-test. Data are provided as Supplementary Data 5. **f** Western blot validations of mitochondrial MARCH5 (ARFGAP1) or centrosomal RNF214 (ROCK1, GIGYF2, and CLINT1) targets identified in €. Arrowheads and bars point to unmodified and Ub modified proteins, respectively. IN: input; St PD: streptavidin pull-down. Molecular weight markers are shown to the left of the blots in kDa. **a**–**f** All BioE3 experiments were performed as above, with biotin supplementation at 50 μM for 2 h (or indicated time points). Data are representative of 3 independent transfection experiments with similar results. Source data are provided in the Source data file.

were used for analyses by western and mass spectrometry, and more adherent U2OS cells for microscopy experiments. For all BioE3 experiments, cells were pre-cultured for 24 h in media containing 10% dialyzed FBS (3.5 kDa MWCO; 150 mM NaCl; filter-sterilized) prior to transfections and subsequent DOX induction, and maintained during DOX induction and timed biotin labelings. Cultured cells were maintained for maximum 20 passages maximum and tested negative for mycoplasma.

### Cloning
All constructs were generated by standard cloning or by Gibson Assembly (NEBuilder HiFi Assembly, NEB) using XL10-Gold bacteria (Agilent). Depending on the construction, plasmid backbones derived from EYFP-N1 (Clontech/Takara), Lenti-Cas9-blast (a kind gift of F. Zhang; Addgene #52962) or TRIPZ (Open Biosystems/Horizon) were used. BirA and bioUb were obtained from CAG-bioUb[15]. NEDD4^WT^ and NEDD4^3M^ were a kind gift from S. Polo and were previously described[50]. SUMO1, SUMO2, CEP120, RNF4, MIB1, PEX12, MARCH5, and RNF214 ORFs were amplified from hTERT-RPE1 cell cDNA by high-fidelity PCR (Platinum SuperFi DNA Polymerase; Invitrogen). A GSQ linker (GGGSSGGGQISYASRG) was placed between the BirA and E3 ligases. Mutations described in the text were introduced by overlap PCR, Quikchange method (Agilent), or by gene synthesis (IDT; Geneart/Thermo Fisher). Constructions were validated by Sanger sequencing. Details of all constructs are described in Supplementary Data 9, and information about primers used in this study is available in Supplementary Data 10. Sequences/maps of representative constructs are available in the Source Data file. Other cloning details are available upon request.

### Lentiviral transduction
Lentiviral expression constructs were packaged in HEK293FT cells using calcium phosphate transfection of psPAX2 and pMD2.G (kind gifts of D. Trono; Addgene #12260, #12259) and pTAT (kind gift of P. Fortes; for TRIPZ-based vectors). Transfection medium was removed after 12–18 h and replaced with fresh media. Lentiviral supernatants were collected twice (24 h each), pooled, filtered (0.45 μm), supplemented with sterile 8.5% PEG6000, 0.3 M NaCl, and incubated 12–18 h at 4 °C. Lentiviral particles were concentrated by centrifugation (1500 × g, 45 min, 4 °C). Non-concentrated virus was used to transduce HEK293FT and 5x concentrated virus was used for U2OS cells. Drug selection was performed with 1 μg/ml puromycin (ChemCruz).

### Transfections and drug treatments
HEK293FT cells were transfected using calcium phosphate method. U2OS cells were transfected using Effectene Transfection Reagent (Qiagen) or Lipofectamine 3000 (Thermo Fisher). 20 nM of siRNAs (sequence: *GGACAUAUUGCUACCUGUUCUUUAU*[78]) for MIB1 knockdown were transfected using Lipofectamine 3000 (Thermo Fisher). For all BioE3 experiments, cells were pre-cultured for 24 h in 10%

dialyzed FBS-containing media prior to transfections. For stably transduced TRIPZ cell lines, induction with DOX (doxycycline hyclate 1 μg/ml; 24 h; Sigma-Aldrich) was performed prior to biotin treatment (50 μM; Sigma-Aldrich) for the indicated exposure times. MG132 (10 μM; ChemCruz), Bortezomib (400 μM; MedChemExpress), ATO (1 μM; Sigma-Aldrich), PR619 (20 μM; Merck), CaCl$_2$ (2 mM; Sigma-Aldrich) and ionomycin (1 μM; Thermo Fisher) treatments were performed (with or without biotin, depending on the experiment; see Supplementary Note 2) prior to cell lysis or immunostaining at the indicated time-points.

### Western blot analysis
Cells were washed 2x with 1x PBS to remove excess biotin and lysed in highly stringent washing buffer 5 (WB5; 8 M urea, 1% SDS in 1x PBS) supplemented with 1x protease inhibitor cocktail (Roche) and 50 μM NEM. Samples were then sonicated and cleared by centrifugation (25,000 × g, 30 min at room temperature, RT). 10–20 μg of protein were loaded for SDS-PAGE and transferred to nitrocellulose membranes. Blocking was performed in 5% milk in PBT (1x PBS, 0.1% Tween-20). Casein-based blocking solution (Sigma) was used for anti-biotin blots. Primary antibodies were incubated overnight at 4 °C and secondary antibodies for 1 h at RT. Primary antibodies used as follows: Cell Signaling Technology: anti-biotin-HRP (1/1000; Cat#7075 S), anti-alpha-Actinin (1/5000; Cat#6487S), anti-PCM1 (1/1000; Cat#5213S); SinoBiological: anti-BirA (1/1000; Cat#11582-T16); Proteintech: anti-USP9X (1/1000; Cat#55054-1-AP), anti-CEP131 (1/1000; Cat#25735-1-AP), anti-SUMO2/3 (1/1000; Cat#67154-1-Ig), anti-GAPDH (1/5000; Cat#60004-1-Ig), anti-PML (1/1000; Cat#21041-1-AP), anti-ROCK1 (1/1000; Cat#21850-1-AP), anti-GIGYF2 (1/1000; Cat#24790-1-AP), anti-CLINT1 (1/1000; Cat#10470-1-AP), anti-ARFGAP1 (1/1000; Cat#13571-1-AP); GenScript: anti-AviTag (1/1000; Cat#A00674), anti-His tag (1/1000; Cat#A00186S); Sigma-Aldrich: anti-CCT8 (1/1000; Cat#HPA021051), anti-TP53BP2 (1/1000; Cat#HPA021603), anti-MIB1 (1/1000; Cat#M5948); anti-ubiquitin (1/5; ZTA10; generated at IFOM[51]); Jackson ImmunoResearch: anti-Mouse-HRP (1/5000; Cat#115-035-062), anti-Rabbit-HRP (1/5000; Cat#111-035-045). Proteins were detected using Clarity ECL (BioRad) or Super Signal West Femto (ThermoFisher) in an iBright CL1500 imaging system (Thermo Fisher). All uncropped blots are provided within the Source data file.

### Immunostaining and confocal microscopy
U2OS cells were seeded on 11 mm coverslips (25,000 cells per well; 24-well plate). After washing 3 times with 1x PBS, cells were fixed with 4% PFA supplemented with 0.1% Triton X-100 in 1x PBS for 15 min at RT. Then, coverslips were washed 3 times with 1x PBS. Blocking was performed for 30 min at RT in blocking buffer (2% fetal calf serum, 1% BSA in 1x PBS). Primary antibodies were incubated for 1–2 h at 37 °C and cells were washed with 1x PBS 3 times. Primary antibodies used as follows: SinoBiological: anti-BirA (1/500; Cat#11582-T16); Novus Biologicals: anti-BirA (1/200; Cat#NBP2-59939); GenScript: anti-AviTag (1/100; Cat#A00674); Proteintech: anti-PML (1/150; Cat#21041-1-AP);

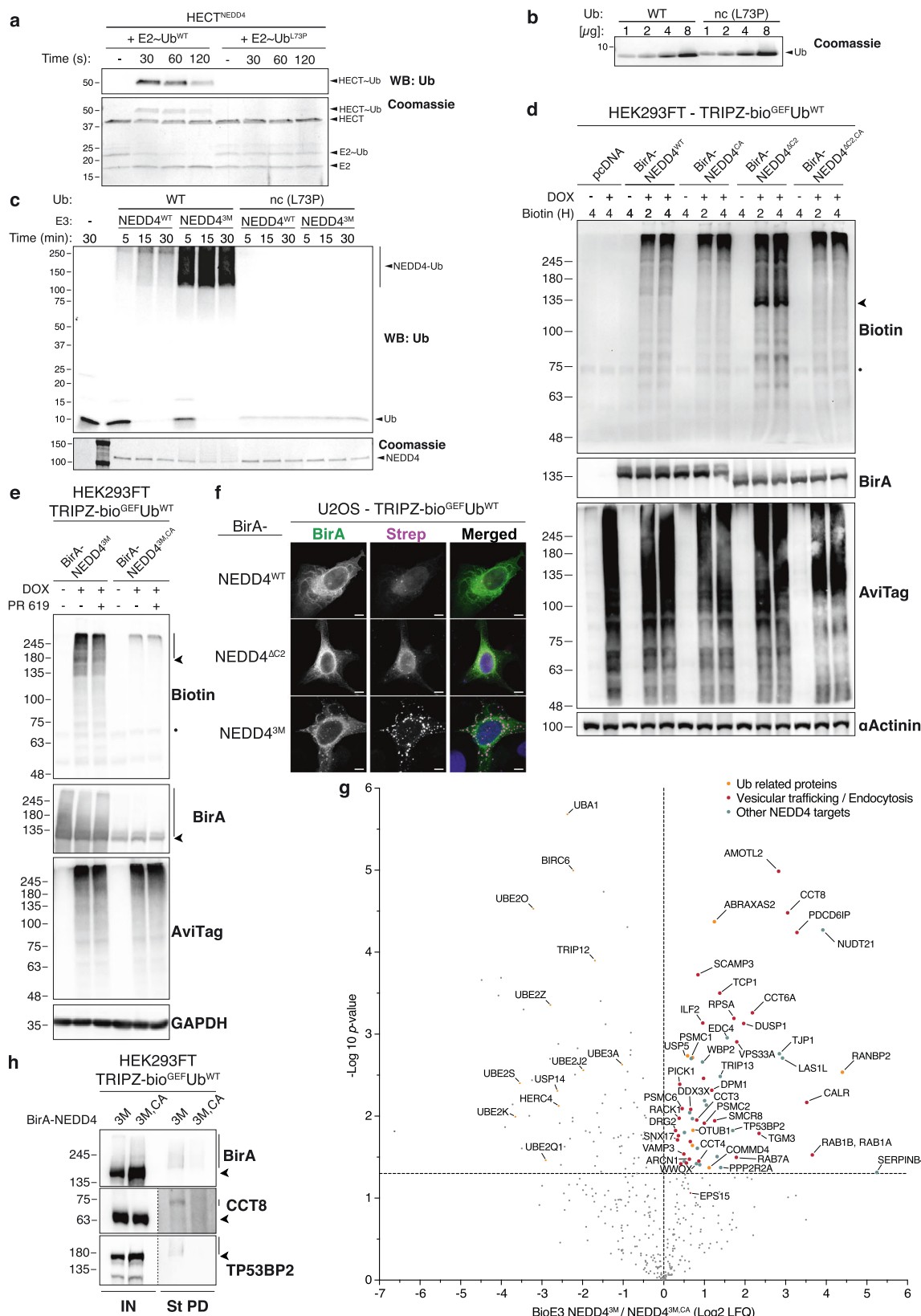

BioLegend: anti-CETN2 (1/100; Cat#698602); BD Biosciences: anti-HSP60 (1/100; Cat#H99020); Secondary antibodies (together with fluorescent streptavidin) were incubated for 1 h at 37 °C, followed by nuclear staining with DAPI (10 min, 300 ng/ml in 1x PBS; Sigma-Aldrich). Secondary antibodies (ThermoFisher) were all used at 1/200: anti-Rabbit Alexa Fluor 488 (Cat#A-11034), anti-Mouse Alexa Fluor 488 (Cat#A-11029), anti-Mouse Alexa Fluor 647 (Cat#A-31571), anti-Rabbit Alexa Fluor 647 (Cat#A-21244), anti-Rat Alexa Fluor 647 (Cat#A-21247). Streptavidin Alexa Fluor 594 (1/200; Cat#016-290-084; Jackson ImmunoResearch) was used. Fluorescence imaging was performed using confocal microscopy (Leica SP8 Lightning) with 63x Plan ApoChromat NA1.4 objective.

**Fig. 8 | BioE3 using bio$^{GEF}$Ub$^{WT}$ identifies targets of activated NEDD4. a–c** Ubnc (L73P) mutation impairs NEDD4-Ub transthiolation and autoubiquitination. **a** Western blot (upper) and Coomassie staining of NEDD4 transthiolation assay, using Ub$^{WT}$ loaded Ube2D3 (E2-Ub$^{WT}$) or Ubnc loaded Ube2D3 (E2 - Ub$^{L73P}$). **b** Coomassie staining showing that Ub$^{WT}$ and Ubnc were at similar levels in the reaction. **c** Western blot (upper) and Coomassie staining of NEDD4 auto-ubiquitination assay using purified Ube1 (E1), Ube2D3 (E2), Ub$^{WT}$ or Ubnc together with NEDD4$^{WT}$ or NEDD4$^{3M}$ (E3s). Ubiquitination reactions were stopped at indicated time-points. NEDD4$^{WT}$ and NEDD4$^{3M}$ autoubiquitination is impaired by L73P mutation on Ub (black arrowhead). Data are representative of 2 independent experiments with similar results. **d, e** Western blot of BioE3 experiments performed on HEK293FT stable cell line expressing TRIPZ-bio$^{GEF}$Ub$^{WT}$ transiently transfected with **d** EFS-BirA-NEDD4$^{WT}$, NEDD4$^{CA}$, NEDD4$^{ΔC2}$ or NEDD4$^{ΔC2,CA}$ and **e** EFS-BirA-NEDD4$^{3M}$ or EFS-BirA-NEDD4$^{3M,CA}$. Active, auto-ubiquitinated and biotinylated BirA-NEDD4$^{ΔC2}$ and BirA-NEDD4$^{3M}$ are depicted with black arrowheads and ubiquitinated substrates with bars. Dots indicate endogenous biotinylated carboxylases. Cells in (**e**) were also treated with the DUB inhibitor PR619. **f** Confocal microscopy of BioE3 experiment performed on U2OS stable cell line for TRIPZ-bio$^{GEF}$Ub$^{WT}$ transfected with EFS-BirA-NEDD4$^{WT}$, BirA-NEDD4$^{ΔC2}$ or BirA-NEDD4$^{3M}$. Biotinylated material is stained with fluorescent streptavidin (Strep, magenta), and BirA with specific antibody (green). Black and white panels show the green and magenta channels individually. Scale bar: 8 µm. **g** Volcano plot of LC-MS analysis comparing streptavidin pull-downs of BioE3 experiments performed on HEK293FT stable cell line expressing TRIPZ-bio$^{GEF}$Ub$^{WT}$ and transfected with EFS-BirA-NEDD4$^{3M}$ or BirA-NEDD4$^{3M,CA}$ ($n = 3$ biological replicates). Proteins significantly enriched (Log2 NEDD4$^{3M}$/NEDD4$^{3M,CA}$ > 0 and $p$-value < 0.05) were considered as NEDD4 targets. Statistical analyses were performed by two-sided Student's $t$-test. Data are provided as Supplementary Data 7. **h** Western blot validations of NEDD4 targets identified in (**g**): CCT8 and TP53BP2. Arrowheads and bars point to unmodified and Ub modified proteins, respectively. IN: input; St PD: Streptavidin pull-down. **a–h** Molecular weight markers are shown to the left of the blots in kDa. **d–h** Data are representative of 3 independent transfection experiments with similar results. Source data are provided in the Source data file.

## Pull-down of biotinylated proteins

Samples were processed as previously described[83]. Briefly, cleared lysates from WB5 lysis buffer were adjusted to the same protein concentration before incubating them with 1/50 (vol$_{beads}$/vol$_{lysate}$) equilibrated NeutrAvidin-agarose beads (ThermoFisher) overnight at RT. Due to the high-affinity interaction between biotin and streptavidin, beads were subjected to stringent series of washes, using the following WBs (vol$_{WB}$/2vol$_{lysate}$): 2x WB1 (8 M urea, 0.25% SDS); 3x WB2 (6 M Guanidine-HCl); 1x WB3 (6.4 M urea, 1 M NaCl, 0.2% SDS); 3x WB4 (4 M urea, 1 M NaCl, 10% isopropanol, 10% ethanol, and 0.2% SDS); 1x WB1; 1x WB5; and 3x WB6 (2% SDS; WB1-6 prepared in 1x PBS). Biotinylated proteins were eluted in 1 vol$_{beads}$ of Elution Buffer (4x Laemmli buffer, 100 mM DTT; 80 µl for LC-MS/MS experiments) through heating at 99 °C for 5 min and subsequent vortexing. Beads were separated using clarifying filters (2000 × $g$, 2 min; Vivaclear Mini, Sartorius).

## Liquid chromatography mass spectrometry (LC-MS/MS)

Stable HEK293FT TRIPZ-bio$^{GEF}$Ubnc or TRIPZ-bio$^{GEF}$Ub$^{WT}$ lines were generated, selected with puromycin (1 µg/ml). Cells were subcloned, and selected clones exhibiting low background and good Dox-inducibility of bioUb were validated by WB and immunostaining prior to use for large-scale mass spectrometry experiments. Unless specified otherwise, the bio$^{GEF}$Ubnc cell line was used. For RNF4 BioE3, cells were transfected with EFS–BirA–RNF4$^{WT}$, EFS–BirA–RNF4$^{CA}$ or EFS–BirA–RNF4$^{ΔSIM}$. For MIB1 BioE3, cells were transfected with EFS–BirA–MIB1$^{WT}$ or EFS–BirA–MIB1$^{CA}$. For MARCH5 and RNF214 BioE3 experiments, cells were transfected with EFS–BirA–MARCH5$^{WT}$, EFS–BirA–MARCH5$^{CA}$, EFS–BirA–RNF214$^{WT}$ or EFS–BirA–RNF214$^{CA}$. For NEDD4 BioE3, the bio$^{GEF}$Ub$^{WT}$ cell line was used, and transfected with EFS–BirA-NEDD4$^{3M}$ or EFS–BirA-NEDD4$^{3M,CA}$. For pilot BioE3 experiments for western analysis and immunofluorescence, controls without DOX induction or biotin labeling were added (except for MARCH5/RNF214).

All mass-spectrometry experiments were performed in triplicates (three independent pull-down experiments). Four confluent 15 cm dishes (=8 × 10$^7$ cells, 2 ml of lysis/plate; 8 ml total) per replicate were analyzed by LC-MS/MS. Samples eluted from the NeutrAvidin beads were separated in SDS-PAGE (50% loaded) and stained with Sypro Ruby (Invitrogen; data provided in the source data file) according to manufacturer's instructions. Gel lanes were sliced into three pieces as accurately as possible to guarantee reproducibility. The slices were subsequently washed in milli-Q water. Reduction and alkylation were performed (10 mM DTT in 50 mM ammonium bicarbonate; 56 °C; 20 min; followed by 50 mM chloroacetamide in 50 mM ammonium bicarbonate; 20 min; protected from light). Gel pieces were dried and incubated with trypsin (12.5 µg/ml in 50 mM ammonium bicarbonate; 20 min; ice-cold). After rehydration, the trypsin supernatant was discarded. Gel pieces were hydrated with 50 mM ammonium bicarbonate, and incubated overnight at 37 °C. After digestion, acidic peptides were cleaned with TFA 0.1% and dried out in a RVC2 25 speedvac concentrator (Christ). Peptides were resuspended in 10 µL 0.1% formic acid (FA) and sonicated for 5 min prior to analysis.

Samples were analyzed using a timsTOF Pro mass spectrometer (trapped ion mobility spectrometry/quadrupole time of flight hybrid; Bruker Daltonics) coupled online to an Evosep ONE liquid chromatography system (Evosep) at the proteomics platform of CIC bioGUNE. This mass spectrometer also uses PASEF scan mode (parallel accumulation – serial fragmentation). Sample (200 ng) was directly loaded in a 15 cm Evosep Endurance C18 column (Evosep) and resolved at 400 nl/min with a 44 min gradient (30 SPD protocol). Column was heated to 50 °C using an oven. Masses were analyzed between 100 and 1700 $m/z$. Mobility was analyzed between 0.6 and 1.6 V·s/cm$^2$ (collision energies for the fragmentation of peptides were 20 eV and 59 eV, respectively), with a ramp and accumulation time of 100 ms. Ten PASEF ramps were established for each cycle. Charges 0–5 were considered in the acquisition.

## Mass spectrometry data analysis

Raw MS files were analyzed using MaxQuant (version 2.2)[84] matching to a human proteome (UP000005640; Uniprot filtered reviewed *H. sapiens* proteome) under default parameters except otherwise stated. A maximum of 2 missed cleavages and precursor and fragment tolerances of 20 ppm (first search) and 10 ppm centroid match tolerance were considered. Cysteine carbamidomethylation was considered as fixed modification, and methionine oxidation as variable. Label-Free Quantification (LFQ) was enabled with default values except for a ratio count set to 1. Slices corresponding to same lanes were considered as fractions. Matching between runs (2 min match window, 20 min alignment match window) was enabled. Only proteins and PSMs identified with FDR < 1% were considered for further analysis. Data were loaded onto the Perseus platform (version 1.6.15)[85] and further processed (Log2 transformation, imputation). Proteins detected with at least 2 peptides and in at least 2 of the 3 replicates in at least one group were included. A two-sided Student's $t$-test was applied to determine the statistical significance of the differences detected ($n = 3$). Data were loaded into GraphPad Prism 8 version 8.4.3 to build the corresponding volcano-plots. All Principal Component Analysis (PCA), correlation Scatter plots and Sypro Ruby gel stainings for each of the LC-MS experiments are provided in the source data file.

Network analysis was performed using the STRING app version 1.4.2[86] in Cytoscape version 3.9.1[87], with a high confidence interaction score (0.7). Transparency and width of the edges were continuously mapped to the String score (text mining, databases, coexpression, experiments, fusion, neighborhood and cooccurrence). The Molecular

COmplex DEtection (MCODE) plug-in version 1.5.1[88] was used to identify highly connected sub-clusters of proteins (degree cutoff of 2; Cluster finding: Haircut; Node score cutoff of 0.2; K-Core of 2; Max. Depth of 100). Gene ontology (GO) analysis was performed using g:Profiler web server (version e108_eg55_p17_0254fbf[89]) and statistical enrichment analysis was performed using Fisher's one-tailed test with g:SCS correction for multiple comparisons. Venn diagrams were drawn using InteractiVenn[90] web tool.

### In vitro transthiolation assay

WT Ub (Sigma) and non-cleavable Ub mutant (Ubnc, L73P, UBPBio) were assayed side by side. E1 and E2 enzymes, and NEDD4 E3 (full-length or HECT domain) were produced in bacteria, as previously described (as GST or 6xHIS fusions, induced and affinity-purified)[51]. These assays were performed in two steps. First, the E1 enzyme (Ube1, 100 nM) was used to load Ub (10 μM; WT or L73P) onto the E2 enzyme (Ube2D3, 5 μM) in ubiquitination buffer (25 mM Tris-HCl, pH 7.6, 5 mM MgCl$_2$, 100 mM NaCl, 2 mM ATP) for 30 min at 37 °C and then quenched on ice by a two-fold dilution with 0.5 M EDTA. Then, the loaded E2 was mixed with HECT$^{NEDD4}$[51] in ubiquitination buffer to the following final concentrations: E2, 1.4 μM; Ub, 2.8 μM; HECT, 1 μM. The reaction mixture was placed at 25 °C, and thioester formation on the HECT$^{NEDD4}$ was monitored by quenching the reaction at different time points with Laemmli buffer without reducing agent, followed by analysis by polyacrylamide gel electrophoresis (SDS-PAGE).

### In vitro ubiquitination assay

Reaction mixtures contained purified enzymes (20 nM E1-Ube1, 250 nM E2-Ube2D3, 250 nM E3), and 1.25 μM of Ub (WT or L73P) in ubiquitination buffer (25 mM Tris-HCl, pH 7.6, 5 mM MgCl$_2$, 100 mM NaCl, 2 mM ATP). Reactions were incubated at 37 °C. At the indicated time point, the reaction mix was stopped by addition of Laemmli buffer with reducing agent (100 mM DTT) before SDS-PAGE analysis. Ubiquitination activity of WT NEDD4 (NEDD4$^{WT}$) was compared with NEDD4 C2-HECT binding surface triple mutant (NEDD4$^{3M}$)[50]. Detection was performed by immunoblotting using mouse monoclonal anti-Ub[51] and Coomassie gel-staining.

### MIB1 knock-out cell line generation

Optimal sgRNA sites to target human MIB1 locus were selected using CRISPOR webtool (MIB1 sgRNA#1: CACTTCCCGGTGTAGTAATT; sgRNA#2: GATGGAGGAAATGGACGTAG; 3 kb deletion predicted). LentiCRISPRv2-blast (kind gifts of B. Stringer; Addgene #98293) was digested with Esp3I/BsmB1 (Thermo) and ligated with corresponding duplex oligos to construct MIB1 targeting vectors (cloning details available upon request). The vectors were packaged into lentiviral particles as described above, and used to transduce HEK293FT cells. Blasticidin selection (5 μg/ml) was applied after 48hrs to select a stable population. After low-density plating and single-cell cloning, 18 clones were screened by western using MIB1 antibody. Several clones with no detection on MIB1 signal were propagated and frozen. Clone#12 was used for TUBEs analysis.

### Purification of ubiquitinated proteins using TUBEs and 6xHis-Ub pull-down

MIB1 KO and siMIB1 cells were lysed in TUBEs lysis buffer and the total ubiquitome was purified using TUBEs as previously described[91]. Briefly, GST-TUBES (ubiquilin-type; gift from M. Rodriguez, LCC-Toulouse) were loaded onto glutathione resin, then incubated with cell lysates (precleared with GST-only resin), washed extensively, and then processed for PAGE. For orthogonal validations, HEK293FT cells were co-transfected with the indicated constructs and pcDNA3-6xHis-Ub (gift from M. Rodriguez, LCC-Toulouse), lysed in lysis buffer (8 M urea, 0.1 M Na$_2$HPO$_4$/NaH$_2$PO$_4$, pH 8.0, 0.01 M Tris-HCl pH 8.0, 10 mM imidazole pH 8.0, 5 mM β-mercaptoethanol, and 0.1% Triton X-100), supplemented

with 1x protease inhibitor cocktail (Roche) and 50 mM NEM. Samples were then sonicated and cleared by centrifugation ($25,000 \times g$, 30 min at RT). Cleared lysates were adjusted to the same protein concentration before incubating them with 1/50 (vol$_{beads}$/vol$_{lysate}$) equilibrated Ni-NTA Agarose beads (Invitrogen) for 2 h at RT. Beads were then washed using WBA (8 M urea, 0.1 M Na$_2$HPO$_4$/NaH$_2$PO$_4$, pH 8.0, 0.01 M Tris-HCl pH 8.0, 10 mM imidazole pH 8.0, 2.5 mM β-mercaptoethanol, and 0.2% Triton X-100), WBB (8 M urea, 0.1 M Na$_2$HPO$_4$/NaH$_2$PO$_4$, pH 6.3, 0.01 M Tris-HCl pH 6.3, 10 mM imidazole pH 7.0, 2.5 mM β-mercaptoethanol, and 0.2% Triton X-100) and WBC (8 M urea, 0.1 M Na$_2$HPO$_4$/NaH$_2$PO$_4$, pH 6.3, 0.01 M Tris-HCl, pH 6.3, no imidazole, 2.5 mM β-mercaptoethanol, and 0.2% Triton X-100). Proteins were eluted with 1 vol$_{beads}$ of Elution Buffer (2x Laemmli buffer, 7 M urea, 0.1 M NaH$_2$PO$_4$/Na$_2$HPO$_4$, 0.01 M Tris/HCl, pH 7.0, and 500 mM imidazole pH 7.0) through heating at 99 °C for 5 min and subsequent vortexing.

### Reporting summary

Further information on research design is available in the Nature Portfolio Reporting Summary linked to this article.

### Data availability

All data supporting the findings are provided within the paper, the Supplementary data, the Supplementary Information and the Source data file. The fasta file of the human proteome (Uniprot filtered reviewed *H. sapiens* proteome) UP000005640 was downloaded from Uniprot. In addition, the mass spectrometry proteomics raw data have been deposited to the ProteomeXchange Consortium via the PRIDE partner repository[92] with the dataset identifier PXD041685. Processed LC-MS/MS data as well as their corresponding gene ontology source data are provided as Supplementary data files. The different datasets used for comparisons in this study are available: SUMOylated protein dataset, https://doi.org/10.1038/nrm.2016.81 [https://www.nature.com/articles/nrm.2016.81]; MIB1 interactome dataset, https://doi.org/10.1038/s41598-019-48902-x [https://www.nature.com/articles/s41598-019-48902-x]; Mitocarta dataset, https://doi.org/10.1093/nar/gkaa1011 [https://www.broadinstitute.org/mitocarta/mitocarta30-inventory-mammalian-mitochondrial-proteins-and-pathways]; mitochondrial interactome dataset, https://doi.org/10.1016/j.cmet.2020.07.017 [https://www.cell.com/cell-metabolism/fulltext/S1550-4131(20)30412-5?_returnURL=https%3A%2F%2Flinkinghub.elsevier.com%2Fretrieve%2Fpii%2FS1550413120304125%3Fshowall%3Dtrue]; and the RNF214 interactome dataset, https://doi.org/10.1016/j.molcel.2017.12.020 [https://www.cell.com/molecular-cell/fulltext/S1097-2765(17)30977-2?_returnURL=https%3A%2F%2Flinkinghub.elsevier.com%2Fretrieve%2Fpii%2FS1097276517309772%3Fshowall%3Dtrue]. Source data are provided with this paper.

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

## Acknowledgements

For scientific support and advice, we thank Iraide Escobes (Proteomics Platform, CIC bioGUNE), Carolina da Fonseca and Arantza Juanes (CIC bioGUNE), Arnoud de Ru and P.A. van Veelen (Center for Proteomics and Metabolomics, LUMC), and Christian Renz (IMB, Mainz). O.B.-G., F.T., R.B. and A.C.O.V. acknowledge funding by the grant 765445-EU (UbiCODE Program). R.B., J.D.S., S.P. and A.C.O.V. acknowledge networking support from the ProteoCure COST Action (CA20113). O.B.-G. acknowledges funding by the FEBS Short-Term Fellowship. R.B. acknowledges MCIN/AEI/10.13039/501100011033 (PID2020-114178GB-I00, SEV-2016-0644, and CEX2021-001136-S Severo Ochoa Excellence Program).

Additional support was provided by the Department of Industry, Tourism, and Trade of the Basque Country Government (Elkartek Research Programs) and by the Innovation Technology Department of the Bizkaia County. L.M.-C. acknowledges FPU grant FPU20/05282 (Ministerio de Educación y Formación Profesional). V.M. acknowledges FPI grant PRE2018-086230 (MCIU/AEI/FEDER, EU). F.E. acknowledges ProteoRed-ISCIII (PT13/0001/0027) and CIBERehd. U.M. acknowledges the Basque Government Department of Education (IT1473-22) and the Spanish MCIU (PID2020-117333GB-I00 (FEDER/EU)).

## Author contributions

O.B.-G., L.M.-C., J.D.S. and R.B. designed experiments, analyzed data and wrote the manuscript. O.B.-G., L.M.-C., V.M., C.P., F.T., V.T., E.M., M.A., I.I. and J.D.S. developed experimental protocols and performed experiments. F.E., U.M., S.P. and A.C.O.V. provided scientific resources.

## Competing interests

The authors declare no competing interests.
