## [Peer Review File · Nature Communications]

REVIEWER COMMENTS

Reviewer #1 (Remarks to the Author):

Barroso-Gomila et al. put forth BioE3, an innovative methodology geared towards the identification of substrates of ubiquitin-like (UbL) E3. The core objective of this study is to distinguish genuine targets of E3s from proteins that interact with E3s. The authors demonstrated the efficacy of BioE3 in recognizing targets using a selection of RING-type E3 ligases, membrane-localized specific E3 ubiquitin ligase, and HECT E3s.

Fig. 2c- the authors utilize confocal microscopy to observe U2OS cell lines that stably express BirA-E3s, as well as the Dox induction of bioUbnc. The negative controls demonstrated in this scenario are 'No Dox+Biotin' and 'Dox+no Biotin'. However, an important additional control could be a stable U2OS cell line expressing an Empty vector of BirA, which would have similar localization to the E3 ubiquitin ligase of interest. In other words, BirA-NLS (Nuclear Localization Signal) and BirA-NES (Nuclear Export Signal) should be considered as controls for BirA-RNF4 and BirA-MIB1, respectively. By adopting this approach, the authors could confirm the cellular localization of the E3 ubiquitin ligases under consideration, as well as verify that the addition of Dox+Biotin to the BirA-[Empty vectors] results in an absence of streptavidin-specific labeling when compared to the BirA-E3s, as anticipated.

Supplementary Figure 1 presents the only comparison between a BirA fused protein and BirA-EV, but in this case, the biotinylation was entirely nonspecific (attributable to BiowheUbnc usage) irrespective of the pulse times employed. Therefore, it does not serve as a viable control for the experiment described in Figure 2c.

Fig 3-5. The authors focus on the BioE3 application for RNF4 (STUbL E3 ubiquitin ligase). They suggest that employing BioE3 enabled the identification of specific ubiquitylated or SUMOylated substrates of RNF4. However, Figure 4, which illustrates volcano plots (4a-b) and a Venn diagram of SUMOylated targets (4d), does not indicate any significant difference between the catalytically inactive form and the delta SIM variant of BirA-RNF4 regarding the candidate substrates. In addition, Figures 4A and 4B do not show the anticipated decrease in UbL and PML protein enrichment, respectively. This pattern actually exposes a methodological limitation in detecting specific bona fide substrates for the E3 ubiquitin ligase's catalytic activity. A potential enrichment could be found by comparing a double mutant (CA and delta SIM) to the single mutants. It would be more appropriate and pertinent to use bioGEF-SUMO1/2nc instead of Ubnc in analyzing STUbL as the authors confirmed the specificity of these tools as shown in Figure 2B.

Fig 7. The authors depict the analysis of BirA-MARCH5 or RNF214 ubiquitin targets. They state that MARCH5 was fused to an N-tagged BirA. However, MARCH5 features a distinct ubiquitin ligase RING domain at the N-terminus and four TM α -helices at the C-terminus. The choice to fuse all E3 ubiquitin ligase candidates with an N-terminal is not an inconsequential one and should be supported by either literature or empirical evidence. It is widely accepted that the functionality of most E3 ligases could be compromised by C-terminal extensions due to their C-terminus catalytic domain¹. However, there are instances, such as with membranal E3s or the Tripartite motif assembly domain (TRIM E3s), where N-terminal tagging might be misleading and potentially interfere with their catalytic activity^{2,3}. Moreover, the enrichment of MARCH5 ubiquitylated targets discovered in this analysis (Fig. 7e) was very specific for mitochondrial proteins. Notably, known substrates such as MFN2 and MCL1 did not achieve statistical significance. Furthermore, a recent paper has illuminated a novel role of MARCH5 in peroxisomal pexophagy, employing another proximity labeling method, 'PUP-IT' (pupylation based interaction tagging), which uses a C-terminal tag³.

Fig 8 delves into the analysis of BirA NEDD4 (HECT-type E3 ligases) targets. The authors endeavored to demonstrate the applicability of BioE3 to identify targets of HECT E3s, but the example of NEDD4 primarily highlighted the limitations of the technique. The system required significant modifications, including the usage of several variants of NEDD4 (auto-inhibited vs. active). The employment of WT-Ub to facilitate transthiolation came at the expense of recycling biotinylated WT Ub to sites distinct from where BirA is located. Indeed, many of the protein 'hits' found to be enriched in this analysis are chaperones, and their role as bona fide ubiquitylation targets of NEDD4 is debatable. The enrichment of the TRiC chaperonin in the NEDD active network most likely resulted from the BirA-NEDD4(3M) variant exhibiting a higher expression level compared to the NEDD4(3M,CA), as indicated in the raw data (Supplementary Data 7- BirA NEDD4(3M) with 'Student's T-test Test statistic NEDD4(3M)_NEDD4(3M,CA)' 2.36- approaching significance).

In conclusion, the task of identifying substrates for a particular E3 and distinguishing between non-covalent interactors versus bona fide targets remains a complex challenge. The novel BioE3 method is intriguing and may prove pertinent and applicable in certain instances of E3 ubiquitin ligase-interactome, although not necessarily for specific ubiquitylation targets. It is also crucial for the authors to address the significance of selecting between N- vs C-terminus fusion of BirA. This decision should be made with careful consideration of the E3 ubiquitin ligase structure, particularly in terms of the catalytic domain and localization sequences.

References:

1 Salvat, C., Wang, G., Dastur, A., Lyon, N. & Huibregtse, J. M. The -4 phenylalanine is required for substrate ubiquitination catalyzed by HECT ubiquitin ligases. *J Biol Chem* 279, 18935-18943 (2004). <https://doi.org/10.1074/jbc.M312201200>

2 Watanabe, M. et al. A substrate-trapping strategy to find E3 ubiquitin ligase substrates identifies Parkin and TRIM28 targets. *Commun Biol* 3, 592 (2020). <https://doi.org/10.1038/s42003-020-01328-y>

3 Zheng, J., Chen, X., Liu, Q., Zhong, G. & Zhuang, M. Ubiquitin ligase MARCH5 localizes to peroxisomes to regulate pexophagy. *J Cell Biol* 221 (2022). <https://doi.org:10.1083/jcb.202103156>

Reviewer #2 (Remarks to the Author):

In the study “BioE3 enables the identification of bona fide targets of E3 ligases” Barroso-Gomila and colleagues showed the use of an improved technique to identify ubiquitylated substrates of E3 ligases. Here they demonstrate that by creating BirA-E3-fusion ligases and expressing these in inducible stable cells with BioGEFUb they could identify substrates of RING-type E3 ligases (RNF4 and MIB1), membrane-localised (MARCH5 and RNF214) and HECT-type (NEDD4) ubiquitin ligases.

Overall, this is a very well-designed study with potential applications for screening for novel ligase-substrate interactions whilst also having the important application of being able to identify targets of ubiquitin ligases that are recruited by PROTACs. This will assist in understanding the off-target effects of PROTAC technology.

The study uses artificial expression of BioGEFUb to be used as the ubiquitin moiety to label the potential substrates and I strongly feel that a traditional standard *in vitro/in vivo* ubiquitylation assay is needed to confirm some of the novel substrates identified (as an orthogonal validation approach). Some of the comments below should be considered in the revised manuscript.

General comments:

For all figures, could you please add arrows on the WB images to indicate the band/s you wish to draw attention to?

Can the BioE3 method be used for E3 ligases that require multi-protein complexes such as SCF, BTB, and APC complexes? RING-type and HECT-type are single protein E3 ligases – is this restricted on size or will multi-complexed E3 ligases require the expression of multiple components for this approach to work?

Please add a comment on how the alteration of the ligase with BirA may alter the enzymatic activity of the ligase, thus for this to work, the enzymatic activity of the ligase must be evaluated first.

In some of the conclusions where potential proteins are substrates of specific E3 ligases, these assays rely on the fact that an artificial BioGEFUb is expressed. Without an orthogonal assay to demonstrate that WT ubiquitin is attached to these substrates, the claim that they are substrates is premature. I agree that the pulldowns in the assay demonstrates that the E3 ligase and the potential substrates in Figures 6-8 are protein-protein interaction partners, but additional assays are required to show that they are substrates (i.e. E3 ligase attaching ubiquitin [not BioGEFUb] on) for some of these is needed to give confidence to readers that the BioE3 is highly selective.

In some areas such as line 261, confirmed ubiquitylation substrates of MIB1 should be validated by an orthogonal approach using normal ubiquitin and/or reverse IPS and/or identification of ubiquitylate site. Despite showing the efficiency of bioGEFUbnc in earlier figures it would be good to convince a reader that independent validation outside of the constructs should be carried out. I realise that in vitro ubiquitylation assay was used to assess the autoubiquitylation status of NEDD4 later on, it would be good to also show that a E3 ligase of interest can ubiquitylate a putative substrate identified by BioE3 in addition to showing protein-protein interaction.

In the volcano plots for the MS data is there any particularly reason why only -log p-value was used as a filtering criteria and not fold change as well? Presumably when comparing wild-type vs catalytically inactive or substrate inactive you would be looking for an enrichment (fold change) potential substrate to give higher confidence that the protein hits are real (e.g., >2-fold and p-value cut-off).

Is there information on the ubiquitin chain linkage information for the BioGEFUb that this technique has a preference for? E.g. BioGEF-Ub preferences for mono- or poly-ubiquitin or is it fairly heterogenous ubiquitin groups that get ubiquitylated – is there a way to compare BioGEFUb to overexpressed Ub to see if these ubiquitylate sites or form polyubiquitin linkages differently. As this may help provide information as to which substrates are preferentially tagged, and which are being directed to specific areas of degradation (e.g. K48 and K63) or locations (K11, K23, K27) in the cell. This may help support the notion that some proteins are directed to autophagy (as shown with the MIB1) or the proteasome with RNF4.

Can the BioE3 approach be used to identify ubiquitylation/SUMO site information? i.e. was there any identified in this study that can be added to support its utility for ubiquitylomics/SUMOylomics (not just on the protein level). If so, how does this approach compare to the standard di-Gly IP approach? SUMOylation site information will require a different enzyme to cleave as it does not give the standard GG- motif.

When assessing all the substrates for the specific E3 ligases used in this work, is there a consensus motif/sequence that the E3's use to bind and ubiquitylate? If you align the sequences amongst the different substrates does it give information on a peptide region that all substrates have in common?

E.g. the well-studied B-TRCP E3 ligase has a preference for substrates with a phosphorylated DpSG motif. This may provide information on the mechanistic information and aide in developing TPD strategies such as PROTACs.

Can the BioE3 approach be benchmarked against a standard E3 ligase IP-MS approach instead of comparing to existing data? As different mass specs and set ups can change the results – it would be good to show in house that the BioE3 approach has an advantage or is complementary to a standard E3 IP-MS set up.

Comments referring to specific figures:

Figure 1: Currently written as:

HEK293FT

U2OS

For clarity, could this please be changed to

HEK293FT/

U2OS

Figure 2

Was BirA alone (without attachment to an E3 ligase) ever used as a background control given the amount of leakiness of expression as observed in some Figures?

Could BirA (inactive) also be used as a negative control?

Figure 3

3A. The abundance of BirA is not the same between WT and CA or deltaSIM. Please show that the difference in biotinylation is not due to this form of enzyme being at a higher concentration (eg, densitometry of biotin normalised to BirA* or determine the conditions where the protein quantities are the same). I understand that CA and deltaSIM are meant to be negative controls however, experience with some catalytically inactive enzymes demonstrates that they are not always 100% inactive.

3C. Please indicate which band or smears of bands represent PML. Could these results be quantitated using ImageJ (with added statistics)?

Figure 4

While ubiquitylation and sumoylation substrates were identified, is there any information on the site information of where the BioGEFUb is labelling? Is the lysine or NH₂ on the N-terminus of the potential substrate hindering or competing with biotinylation – will this affect selectivity and site specific information?

Shouldn't the abundance of the BirA*-ligase also be accounted for in your analysis pipeline?

Figure 6

-Because an artificial form of BioGEFUbnc, which is uncleavable by Dubs, is expressed as a single copy in these cell lines, it would be valuable to show that some of these substrates can be modified by endogenous (in vivo) or recombinant (in vitro) Ub or SUMO to demonstrate that these are indeed putative substrates. The BioE3 approach is great for discovery but I think demonstrating 1 or 2 substrates that this is highly selective to identify true substrates is needed.

Figure 7

7F. Please add a lane to the WB figure to show the abundance of each enzyme here that led to these results.

REVIEWER COMMENTS

Reviewer #1 (Remarks to the Author):

Barroso-Gomila et al. put forth BioE3, an innovative methodology geared towards the identification of substrates of ubiquitin-like (Ubl) E3. The core objective of this study is to distinguish genuine targets of E3s from proteins that interact with E3s. The authors demonstrated the efficacy of BioE3 in recognizing targets using a selection of RING-type E3 ligases, membrane-localized specific E3 ubiquitin ligase, and HECT E3s.

We thank the reviewer for the positive comments.

Fig. 2c- the authors utilize confocal microscopy to observe U2OS cell lines that stably express BirA-E3s, as well as the Dox induction of bioUbnc. The negative controls demonstrated in this scenario are 'No Dox+Biotin' and 'Dox+no Biotin'. However, an important additional control could be a stable U2OS cell line expressing an Empty vector of BirA, which would have similar localization to the E3 ubiquitin ligase of interest. In other words, BirA-NLS (Nuclear Localization Signal) and BirA-NES (Nuclear Export Signal) should be considered as controls for BirA-RNF4 and BirA-MIB1, respectively. By adopting this approach, the authors could confirm the cellular localization of the E3 ubiquitin ligases under consideration, as well as verify that the addition of Dox+Biotin to the BirA-[Empty vectors] results in an absence of streptavidin-specific labelling when compared to the BirA-E3s, as anticipated.

The objective of Fig. 2c is to depict the use of bio^{GEF} tag compared to the bio^{WHE} tag (which is commonly used in the bioUb approach), and the specificity obtained. Comparing both tags, we could observe that using the low affinity bio^{GEF} tag the streptavidin signal co-localized well with the BirA signal for both RNF4 and MIB1 E3 ligases, that have very different localization patterns. To further show the specificity of the E3 target labelling, in the following figures corresponding to each ligase, the negative controls used by confocal microscopy were the catalytically inactive (CA) versions (Fig. 3b for RNF4 and Fig. 6b for MIB1). We believe that the CA versions are better negative controls than NLS or NES containing BirAs, as the CA versions display the same localisation than the WT versions (as shown in stainings using anti-BirA antibodies) but they do not generate biotinylated material as the E3 is inactive (Streptavidin panels). While BirA-NLS and BirA-NES could serve as additional controls, we did not have them available. Because these localization tags would not serve to bring BirA in proximity to bio^{GEF}-Ub, we would expect the biotinylation capacity to be negligible (similar to bio^{GEF}-Ub and BirA alone with no tag, analysed by western in Fig 2a).

Supplementary Figure 1 presents the only comparison between a BirA fused protein and BirA-EV, but in this case, the biotinylation was entirely nonspecific (attributable to Bio^{whe}Ubnc usage) irrespective of the pulse times employed. Therefore, it does not serve as a viable control for the experiment described in Figure 2c.

We agree with the Reviewer in that the BirA-EV is not the best negative control for BioE3 experiments. The goal of Supplementary Fig. 1 is to demonstrate the need to use biotin-depleted media (made with dialyzed serum) to obtain the proper control of biotin pulse-labelling. Also, it shows that the WT AviTag (bio^{WHE}), which is commonly used in the bioUbl strategy (Pirone *et al.*, 2017; DOI: 10.1038/srep40756), is not useful for BioE3 strategy due to its general/unspecific labelling. In Fig 2c, we show the gain in spatial specificity of the biotin-labelling of bio^{GEF}Ub when using BirA-RNF4 or BirA-MIB1 compared to the WT bio^{WHE} AviTag (in a sense, we are using them as controls for each other). In the following figures corresponding to each ligase, the negative controls used were the catalytically inactive (CA) versions of the ligases, which we believe are the best negative controls. Although Fig 2c shows IF in U2OS cells, in Fig 2a we used a different technique (western) and different cells (HEK293FT) to show that BirA-EV

+ bio^{GEF}Ub removes the non-specificity seen with BirA-EV + bio^{WHE}Ub. While one can expect how this difference would appear by IF in U2OS, we have not added this extra control, to concentrate on other points.

Fig 3-5. The authors focus on the BioE3 application for RNF4 (STUbL E3 ubiquitin ligase). They suggest that employing BioE3 enabled the identification of specific ubiquitylated or SUMOylated substrates of RNF4. However, Figure 4, which illustrates volcano plots (4a-b) and a Venn diagram of SUMOylated targets (4d), does not indicate any significant difference between the catalytically inactive form and the delta SIM variant of BirA-RNF4 regarding the candidate substrates. In addition, Figures 4A and 4B do not show the anticipated decrease in UbL and PML protein enrichment, respectively. This pattern actually exposes a methodological limitation in detecting specific bona fide substrates for the E3 ubiquitin ligase's catalytic activity. A potential enrichment could be found by comparing a double mutant (CA and delta SIM) to the single mutants. It would be more appropriate and pertinent to use bioGEF-SUMO1/2nc instead of Ubnc in analyzing STUbL as the authors confirmed the specificity of these tools as shown in Figure 2B.

Candidate substrates identified by comparing RNF4 BioE3 to its CA catalytically inactive form or the delta SIM variant point to the fact that RNF4 needs both, the SIMs to recruit polySUMOylated targets, and the RING domain to modify the substrates with Ub (the defining properties of a STUbL). Most RNF4 targets are SUMOylated and subsequently ubiquitinated. As we mentioned in the manuscript, RNF4 "targets were largely dependent on SUMO-SIM interactions, as BirA-RNF4^{ΔSIM} showed biotinylation similar to the background obtained with RNF4^{CA} (Fig. 3a, biotin blot)". Both negative controls (CA and delta SIM) showed decreased levels of labelling of Ub targets of RNF4, and enabled us to identify them. Although we agree that a double mutant (CA and delta SIM) would be a very strong negative control and would likely show an even higher reduction in labelled substrates, we believe that the additional identification of targets comparing RNF4 BioE3 to the double mutant would not provide any new information to the experiments already performed.

The BioE3 technique should only label the ubiquitylated substrates of RNF4, since the biotinylation by BirA only happens when bio^{GEF}-Ub is nearby. Theoretically, the biotinylation could occur as the E2-bio^{GEF}Ub engages BirA-RNF4, or after the bio^{GEF}Ub is transferred to SUMOylated targets. We have shown that the BirA-RNF4-mediated transfer of the biotinylated bio^{GEF}Ub is dependent on both its RING domain (WT vs CA) and its SIMs (WT vs delta SIM), consistent with the fact that RNF4 is a STUbL. Concerning both datasets, we identified SUMO1 and SUMO2 peptides as highly enriched, indicating that the substrates of RNF4 are likely polySUMOylated before being ubiquitinated by RNF4. This is evident also by SUMO2/3 western (Fig 4c; note that RNF4 likely functions as a dimer, and background may arise due to BirA-RNF4mutants complexing with endogenous RNF4). We agree that using bio^{GEF}SUMO1/2nc could be an interesting experiment to identify SUMOylated interactors of RNF4 (comparing RNF4 WT vs delta SIM), but not to identify substrates of RNF4 *per se*, as all SUMOylated proteins interacting with RNF4-SIMs would be labelled, without the requirement for them to be ubiquitylated (although many eventually would be). Thus, we believe that performing BioE3 experiments with bio^{GEF}Ub and comparing the substrates obtained with RNF4 to its delta SIM version, is the best way to show that those are in fact SUMO-dependent substrates of RNF4, as they would require both conditions: 1) the interaction of the SUMO moieties on the substrates with the SIMs of RNF4 and 2) the transfer of the biotinylated bio^{GEF}Ub to the substrate, which does not happen if the SUMO-SIM interaction is impaired.

Fig 7. The authors depict the analysis of BirA-MARCH5 or RNF214 ubiquitin targets. They state that MARCH5 was fused to an N-tagged BirA. However, MARCH5 features a distinct ubiquitin ligase RING domain at the N-terminus and four TM α -helices at the C-terminus. The choice to fuse all E3 ubiquitin ligase candidates with an N-terminal is not an inconsequential one and should be supported by either

literature or empirical evidence. It is widely accepted that the functionality of most E3 ligases could be compromised by C-terminal extensions due to their C-terminus catalytic domain¹. However, there are instances, such as with membranal E3s or the Tripartite motif assembly domain (TRIM E3s), where N-terminal tagging might be misleading and potentially interfere with their catalytic activity.^{2,3} Moreover, the enrichment of MARCH5 ubiquitylated targets discovered in this analysis (Fig. 7e) was very specific for mitochondrial proteins. Notably, known substrates such as MFN2 and MCL1 did not achieve statistical significance. Furthermore, a recent paper has illuminated a novel role of MARCH5 in peroxisomal pexophagy, employing another proximity labelling method, 'PUP-IT' (pupylation based interaction tagging), which uses a C-terminal tag³.

Thanks - this is a good point that we did not address thoroughly in the manuscript. We generally used BirA as an N-terminal fusion with minor consideration towards previous knowledge (structure, other epitope fusions in literature, positioning of the RING domain, etc.) How a fusion behaves can be difficult to predict. As a minimal check, after tagging, we checked subcellular localization using anti-BirA IF to confirm that construct did not totally mislocalize according to known data (e.g. RNF4 to nucleus + nuclear bodies, MIB1 to centriolar satellites, MARCH5 to mitochondria, NEDD4 to membranes/vesicles, etc.) Through our collaborators, we knew that any C-terminal tagging of NEDD4 blocks its catalytic activity (Maspero *et al.*, 2013; DOI: 10.1038/nsmb.2566), so we opted for N-terminal tagging.

We note that a functional FLAG-MARCH5 N-terminal fusion has been reported (Gu *et al.*, 2015; DOI: 10.1038/ncomms8112), but BirA is larger. The reviewer's point regarding MARCH5 was intriguing, and we thought that maybe a C-terminal fusion might yield a more active enzyme that would allow more robust biotinylation of MCL1 and MFN2 (known targets). For this revision, we did attempt to make C-terminally tagged MARCH5-BirA, but two rounds of experiments yielded only out-of-frame clones and for time restraints, unfortunately the question is still unresolved. Even so, we have added text to the manuscript to encourage researchers to consider these issues when choosing type of BirA fusion (N-term, C-term, or even internal) to use. Localization of BirA-fusions can be compared to endogenous localizations using antibodies if available. Also, comparing both N-terminal and C-terminal fusions might yield overlapping sets of candidate substrates, increasing confidence for validation and follow-up.

To stress this important point, we added text in the *Discussion* section, as well as when describing each of the BirA-E3 fusions used in this study in the *Results* section.

Added text in Results:

In reference to RNF4 and MIB1: "BirA was fused at the N-terminus of both ligases to minimize any steric effect on the C-terminal RING domains."

In reference to MARCH5: "BirA was fused at the N-terminus of MARCH5, and even if the RING domain is close to N-terminus, previous N-terminal tagging using FLAG has been reported³⁹."

In reference to RNF214: "BirA was fused at the N-terminus of RNF214 to minimize any steric effect on the C-terminal RING domain."

In reference to NEDD4: "BirA was fused at the N-terminus of NEDD4, as the location of any tag in the C-terminal part abrogates the catalytic activity of the enzyme⁵¹."

Added to Discussion: "While we only tested N-terminal BirA fusions to E3 ligases in this study, we expect that activity and substrates identified could vary depending on linker length and positioning of the BirA-tagging relative to the E3 (N-terminal, C-terminal, or even internal). This should be decided depending on the information available for a particular E3 of interest."

Fig 8 delves into the analysis of BirA NEDD4 (HECT-type E3 ligases) targets. The authors endeavored

to demonstrate the applicability of BioE3 to identify targets of HECT E3s, but the example of NEDD4 primarily highlighted the limitations of the technique. The system required significant modifications, including the usage of several variants of NEDD4 (auto-inhibited vs. active). The employment of WT-Ub to facilitate transthiolation came at the expense of recycling biotinylated WT Ub to sites distinct from where BirA is located. Indeed, many of the protein 'hits' found to be enriched in this analysis are chaperones, and their role as bona fide ubiquitylation targets of NEDD4 is debatable. The enrichment of the TRiC chaperonin in the NEDD active network most likely resulted from the BirA-NEDD4(3M) variant exhibiting a higher expression level compared to the NEDD4(3M,CA), as indicated in the raw data (Supplementary Data 7- BirA NEDD4(3M) with 'Student's T-test Test statistic NEDD4(3M)_NEDD4(3M,CA)' 2.36- approaching significance).

Adapting BioE3 to HECTs took into consideration both the extra transthiolation step and the auto-inhibition/activation of NEDD4 ligase. This example highlights that previous knowledge about the E3 of interest should be considered when available. But rather than posing limits, the NEDD4 experiments showed the specificity, versatility and adaptability of the approach. In fact, NEDD4 BioE3 clearly showed dependence on its Ub-ligase activity, which is the main goal of the BioE3 technique: NEDD4 needs to be activated (either by stimuli such as Ca²⁺, via ionomycin, or using activated mutant NEDD4^{3M}) and to have proper transthiolation of bio^{GEF}Ub to label the substrates. Because of the mutation in bio^{GEF}Ubnc (the “no-cut” low-recycling version) was poorly transferred from E2 to NEDD4, we used wild-type bio^{GEF}Ub. Both versions will be made available to users, but future experiments may use exclusively the wild-type version. We believe that other optimizations (implementation of inducible expression of bio^{GEF}Ub and the use of biotin-depleted media and short biotin-labelling times) limit the bio^{GEF}Ub recycling to levels that don't interfere with analysis.

We agree that detecting components of the TRiC/CCT complex as bona fide targets of NEDD4 could be debatable. The TRiC complex could be recruited if there was misfolding of the NEDD4 activated mutant, although this complex is less linked to stress-induced misfolding (like heat-shock proteins) and selectively associates with ~10% of the proteome, often for co-translational folding (PMID 30978594). Experts on the NEDD4 ligase participating in this study (Simona Polo's lab) have previously characterized the activated NEDD4 versions (Mari *et al.*, 2014; DOI: 10.1016/j.str.2014.09.006). In order to check the correct folding of the NEDD4^{3M} ligase compared to the NEDD4^{WT}, we performed gel filtration assays (see Fig. 1 of this rebuttal letter). Results showed that in fact, the NEDD4^{3M} is properly folded and migrates equally compared to the NEDD4^{WT}, at least in the form of purified protein. Thus, we believe that the proteins identified by NEDD4 BioE3, most of them related to vesicular trafficking, are in fact targets of NEDD4. The gel filtration assay and the purification of the NEDD4^{3M} mutant figure has been included as Supplementary Fig. 10d and e, and text has been included in the *Results* section.

Added text in Results:

“The activated NEDD4^{3M} was confirmed to be properly folded by gel filtration experiments (Supplementary Fig. 10d and e).”

Figure 1: Purification of NEDD4 full-length enzyme. (a) Elution profile on Superdex S200 10/300 of NEDD4^{WT} (blue line), NEDD4^{3M} (yellow line), and gel filtration markers (black dashed line). (b) Coomassie-stained SDS-PAGE gel of the fractions related to the NEDD4^{WT} and NEDD4^{3M} peaks, from 60 to 70 ml. (Note: corresponds to Supp Fig. 10d,e in revised manuscript).

Furthermore, in our efforts to orthogonally validate some targets, we validated TRiC component TCP1 as NEDD4 substrate by performing a GFP-trap experiment (co-transfecting BirA-NEDD4^{3M} or BirA-NEDD4^{3M,CA} together with GFP-TCP1). Results showed that in fact, TCP1 ubiquitination levels increased almost 6 times when co-expressing it with NEDD4^{3M} compared to NEDD4^{3M,CA}, confirming that it is a NEDD4 Ub target (see Fig.2 of this rebuttal letter). In addition, CCT8 was also confirmed as NEDD4 substrate by His⁶-Ub pull down assay (see new Supplementary Fig. 11a for this and other orthogonal validations).

Figure 2: TCP1 is a Ub target of NEDD4. GFP-trap experiment showing the increased ubiquitinated levels of TCP1, when co-expressing GFP-TCP1 together with BirA-NEDD4^{3M} over BirA-NEDD4^{3M,CA}. The Ub blot signal of

purified GFP-TCP1 was quantified, normalised to GFP levels and expressed as fold change over the signal detected in NEDD4^{3M,CA} condition (5.8 higher Ub-TCP1 when expressing NEDD4^{3M} over NEDD4^{3M,CA}).

Regarding NEDD4 levels, evaluation of the levels of BirA-NEDD4^{3M} compared to the negative control NEDD4^{3M,CA} has to be performed in the "INPUT" fraction. Please consider that the blots shown in Figure 8e were performed to test the effectiveness of the CA mutation in the context of the 3M mutant, before scaling up. The samples used for the MS are the ones shown in Figure 8h. To make this point clearer, we included the corresponding BirA blot in Figure 8h, where it can be observed that the levels of BirA are similar, or even higher in the case of the CA mutant (Fig. 3 of this rebuttal letter). This BirA blot has been included and the corresponding Fig. 8h has been updated (see new Fig. 8h).

Figure 3: BirA-NEDD4^{3M} and NEDD4^{3M,CA} expression levels are similar for LC-MS experiment. (Note: corresponds to Fig. 8h in revised manuscript).

Finally, the levels in the pull-down detected by LC-MS (Supplementary data 7, BirA NEDD4(3M)) reflect the auto-ubiquitination of BirA-NEDD4^{3M}, as it has to be modified by biotinylated bio^{GEF}-Ub to be pulled-down after streptavidin purification, and those levels are in fact expected to be higher than those of the transthiolation-deficient NEDD4^{3M,CA} that cannot auto-ubiquitinate.

In conclusion, the task of identifying substrates for a particular E3 and distinguishing between non-covalent interactors versus bona fide targets remains a complex challenge. The novel BioE3 method is intriguing and may prove pertinent and applicable in certain instances of E3 ubiquitin ligase-interactome, although not necessarily for specific ubiquitylation targets. It is also crucial for the authors to address the significance of selecting between N- vs C-terminus fusion of BirA. This decision should be made with careful consideration of the E3 ubiquitin ligase structure, particularly in terms of the catalytic domain and localization sequences.

As with any protein fusion experiment, we fully agree with the Reviewer that the decision on where to fuse BirA (N-terminal, C-terminal, or internal) should be carefully made, considering knowledge available for each particular E3. Both N- and C-terminal fusions would also be informative, especially if working with large or poorly-studied E3 ligase. As mentioned above, we have included new text throughout manuscript to emphasize this point.

Regarding the other point, we strongly believe that BioE3 is enriching for *bona fide* substrates rather than a simple interactome of E3 ligases. There are other techniques that can be used to identify the direct or proximal interactomes of a given protein (pull-downs/coimmunoprecipitations or proximity assays using BioID/TurboID/APEX2, followed by mass spectrometry). While many interactors

identified by these methods may also be BioE3 substrates, there may be substrates that interact too weakly or transiently to be identified as interactors using pulldowns/coimmunoprecipitations. Also, there may be E3 interactors (e.g. necessary for localization, trafficking, scaffolding into complexes) that are not necessarily BioE3 substrates. A thorough exploration of a single E3 using multiple strategies (Co-IP, TurboID, BioE3, differential tagging, etc – all followed by mass spec analysis) would be informative to explore these differences, but beyond the scope and objective of this study. To reiterate, BioE3 identifies *bona fide* targets of E3s, which may/may not be defined as interactors, depending on the additional methods used. For all experiments, we select only those targets that are significantly more abundant in the E3-WT than in the E3-CA. All optimizations were geared to increase specificity of the system. BioE3 yields candidate substrates, with follow-up validations and biology experiments necessary to understand roles for the E3-substrate relationship.

References:

- 1 Salvat, C., Wang, G., Dastur, A., Lyon, N. & Huijbregtse, J. M. The -4 phenylalanine is required for substrate ubiquitination catalyzed by HECT ubiquitin ligases. *J Biol Chem* 279, 18935-18943 (2004). <https://doi.org/10.1074/jbc.M312201200>
- 2 Watanabe, M. et al. A substrate-trapping strategy to find E3 ubiquitin ligase substrates identifies Parkin and TRIM28 targets. *Commun Biol* 3, 592 (2020). <https://doi.org/10.1038/s42003-020-01328-y>
- 3 Zheng, J., Chen, X., Liu, Q., Zhong, G. & Zhuang, M. Ubiquitin ligase MARCH5 localizes to peroxisomes to regulate pexophagy. *J Cell Biol* 221 (2022). <https://doi.org/10.1083/jcb.202103156>

Reviewer #2 (Remarks to the Author):

In the study “BioE3 enables the identification of bona fide targets of E3 ligases” Barroso-Gomila and colleagues showed the use of an improved technique to identify ubiquitylated substrates of E3 ligases. Here they demonstrate that by creating BirA-E3-fusion ligases and expressing these in inducible stable cells with BioGEFUb they could identify substrates of RING-type E3 ligases (RNF4 and MIB1), membrane-localised (MARCH5 and RNF214) and HECT-type (NEDD4) ubiquitin ligases.

Overall, this is a very well-designed study with potential applications for screening for novel ligase-substrate interactions whilst also having the important application of being able to identify targets of ubiquitin ligases that are recruited by PROTACs. This will assist in understanding the off-target effects of PROTAC technology.

The study uses artificial expression of Bio^{GEF}Ub to be used as the ubiquitin moiety to label the potential substrates and I strongly feel that a traditional standard in vitro/in vivo ubiquitylation assay is needed to confirm some of the novel substrates identified (as an orthogonal validation approach). Some of the comments below should be considered in the revised manuscript.

We thank the reviewer for the positive comments. We agree that orthogonal validation of substrates would strongly support the conclusions of this manuscript, and in this revision, we have provided data in this direction.

General comments:

For all figures, could you please add arrows on the WB images to indicate the band/s you wish to draw attention to?

Arrows and bars in WB to indicate the band/s to draw attention to have been included in the figures.

Can the BioE3 method be used for E3 ligases that require multi-protein complexes such as SCF, BTB, and APC complexes? RING-type and HECT-type are single protein E3 ligases – is this restricted on size or will multi-complexed E3 ligases require the expression of multiple components for this approach to work?

Here we show the application of BioE3 strategy to study the substrates of RING and HECT type E3s. It would be very interesting to apply BioE3 in other scenarios, including multi-complex E3s such as Cullin-RING ligases (CRLs). We also envision that BioE3 could be further extended by fusing BirA to cullin scaffolds, adaptors, E2s, ubiquitin-binding proteins etc. (any enzymes/proteins that are modified or proximal to Ub) and perform experiments in TRIPZ-bio^{GEF}Ub^{WT} cell lines, although proof of principle experiments are needed. This exploration is beyond the scope of this manuscript. We note that a recently posted preprint uses a similar approach to BioE3 (called Ub-POD) to query substrates of Rad18, TRAF6, and CHIP/STUB1 (a U-box-containing E3 ligase), supporting that these biotin-based approaches are robust and flexible for E3-substrate discovery.

(Bhogaraju et al. 2023; <https://doi.org/10.1101/2023.09.04.556194>)

To highlight the potential of BioE3 for other E3s, we added a phrase in the *Discussion* section.

Text added: “BioE3 could further be applied to multi-protein complex E3s (e.g. APC/C, SCF, other Cullin RING ligases) to explore substrate specificity,”

Please add a comment on how the alteration of the ligase with BirA may alter the enzymatic activity of the ligase, thus for this to work, the enzymatic activity of the ligase must be evaluated first.

Thanks - this is a good point that we did not address thoroughly in the manuscript. Because both reviewers made the same point, we copy below our comments addressing this point.

Note: Regarding the testing of enzymatic activity, if a substrate and appropriate assay is available for such testing, it could be performed. Also, an auto-ubiquitination test could be done, which works for some E3s. We have not tried, but anti-BirA antibody may be able to immunoprecipitate, to help facilitate such assays. However, many labs are not equipped to do in vitro ubiquitination assays and they can be challenging to set up.

We generally used BirA as an N-terminal fusion with minor consideration towards previous knowledge (structure, other epitope fusions in literature, positioning of the RING domain, etc.) How a fusion behaves can be difficult to predict. As a minimal check, after tagging, we checked subcellular localization using anti-BirA IF to confirm that construct did not totally mislocalize according to known data (e.g. RNF4 to nucleus + nuclear bodies, MIB1 to centriolar satellites, MARCH5 to mitochondria, NEDD4 to membranes/vesicles, etc.) Through our collaborators, we knew that any C-terminal tagging of NEDD4 blocks its catalytic activity (Maspero *et al.*, 2013; DOI: 10.1038/nsmb.2566), so we opted for N-terminal tagging.

We note that a functional FLAG-MARCH5 N-terminal fusion has been reported (Gu *et al.*, 2015; DOI: 10.1038/ncomms8112), but BirA is larger. The reviewer’s point regarding MARCH5 was intriguing, and we thought that maybe a C-terminal fusion might yield a more active enzyme that would allow

more robust biotinylation of MCL1 and MFN2 (known targets). For this revision, we did attempt to make C-terminally tagged MARCH5-BirA, but two rounds of experiments yielded only out-of-frame clones and for time restraints, unfortunately the question is still unresolved. Even so, we have added text to the manuscript to encourage researchers to consider these issues when choosing type of BirA fusion (N-term, C-term, or even internal) to use. Localization of BirA-fusions can be compared to endogenous localizations using antibodies if available. Also, comparing both N-terminal and C-terminal fusions might yield overlapping sets of candidate substrates, increasing confidence for validation and follow-up.

To stress this important point, we added text in the *Discussion* section, as well as when describing each of the BirA-E3 fusions used in this study in the *Results* section.

Added text in Results:

In reference to RNF4 and MIB1: "BirA was fused at the N-terminus of both ligases to minimize any steric effect on the C-terminal RING domains."

In reference to MARCH5: "BirA was fused at the N-terminus of MARCH5, and even if the RING domain is close to N-terminus, previous N-terminal tagging using FLAG has been reported³⁹."

In reference to RNF214: "BirA was fused at the N-terminus of RNF214 to minimize any steric effect on the C-terminal RING domain."

In reference to NEDD4: "BirA was fused at the N-terminus of NEDD4, as the location of any tag in the C-terminal part abrogates the catalytic activity of the enzyme⁵¹."

Added to Discussion: "While we only tested N-terminal BirA fusions to E3 ligases in this study, we expect that activity and substrates identified could vary depending on linker length and positioning of the BirA-tagging relative to the E3 (N-terminal, C-terminal, or even internal). This should be decided depending on the information available for a particular E3 of interest."

In some of the conclusions where potential proteins are substrates of specific E3 ligases, these assays rely on the fact that an artificial Bio^{GEF}Ub is expressed. Without an orthogonal assay to demonstrate that WT ubiquitin is attached to these substrates, the claim that they are substrates is premature. I agree that the pulldowns in the assay demonstrates that the E3 ligase and the potential substrates in Figures 6-8 are protein-protein interaction partners, but additional assays are required to show that they are substrates (i.e. E3 ligase attaching ubiquitin [not Bio^{GEF}Ub] on) for some of these is needed to give confidence to readers that the BioE3 is highly selective.

In some areas such as line 261, confirmed ubiquitylation substrates of MIB1 should be validated by an orthogonal approach using normal ubiquitin and/or reverse IPS and/or identification of ubiquitylate site. Despite showing the efficiency of bio^{GEF}Ubnc in earlier figures it would be good to convince a reader that independent validation outside of the constructs should be carried out. I realise that in vitro ubiquitylation assay was used to assess the autoubiquitylation status of NEDD4 later on, it would be good to also show that a E3 ligase of interest can ubiquitylate a putative substrate identified by BioE3 in addition to showing protein-protein interaction.

We agree with the Reviewer that using orthogonal approaches to validate the substrates would reinforce the conclusions of the manuscript. In this revision, we aimed to validate different targets identified for all the E3s used in this study (compiled in new Supplementary Fig. 11).

Using a common tool for enriching ubiquitylated proteins (His⁶-Ub), we performed His⁶-Ub pull-down experiments expressing different RING ligases (BirA-RNF4, MIB1, MARCH5, RNF214 and NEDD4) and could confirm several hits as targets since their ubiquitinated levels were higher using wild-type versions compared to their respective CA versions (see Supp Fig. 11a).

Also, we generated MIB1 *knock out* (MIB1-KO) cells and performed an enrichment of the endogenous ubiquitome using TUBEs (Mattern *et al.*, 2019; DOI: 10.1016/j.tibs.2019.01.011). In parallel, we also performed a TUBEs experiment by *knocking down* MIB1 using siRNAs. Cells were also treated or not with the proteasomal inhibitor bortezomib to further enrich the ubiquitinated fraction. Using these approaches, we could validate CEP131 and USP9X as Ub targets of MIB1, as the endogenously ubiquitinated fractions of those hits were reduced when *knocking out* or *knocking down* MIB1 (see Supp Fig. 11b and c, respectively).

In the case of NEDD4, we also performed an *in-vitro* ubiquitination assay using purified WBP2 as a substrate. We confirmed WBP2 as a direct target of NEDD4, with no ubiquitination observed using the transthiolation mutant NEDD4^{CS} (see Fig. 7d and e of this rebuttal letter). We note that WBP2 was also previously described as NEDD4 substrate (Kim *et al.* 2009; DOI: 10.1128/MCB.00240-09).

In summary, we used diverse methodologies (GFP-trap pulldowns [GFP-TCP1], His⁶-Ub, TUBEs, *in vitro*) to verify the ubiquitination of substrates of all the E3 ligases under study, using diverse negative controls as CA mutants, KOs and KDs. These orthogonal validations have been added as Supplementary Fig. 11. Comments on these validations were included in the main text, *Results* section, and the *Methods* section has been updated.

Added text in Results:

“Orthogonal validations of BioE3 targets

*To further support that targets were indeed ubiquitylated by the different E3 ligases that we studied, we used orthogonal approaches for validation. We performed His⁶-Ub pulldown experiments, comparing wild-type and catalytic mutant versions of the different BirA-E3s used in this study to confirm ubiquitination status of selected hits (e.g. SUMO-conjugates:RNF4; USP9X/CEP131:MIB1;ARFGAP1:MARCH5; ROCK1:RNF214; CCT8/TP53BP2:NEDD4-3M;Supplementary Fig. 11a). In addition, we generated MIB1 knockout (MIB1-KO) cells, or depleted MIB1 using RNAi, and then enriched the endogenous ubiquitome using TUBEs, with and without proteasome inhibition by bortezomib (Mattern *et al.*, 2019; DOI: 10.1016/j.tibs.2019.01.011). Using this approach, we could validate CEP131 and USP9X as Ub targets of MIB1, as the endogenously ubiquitinated fraction of those hits were reduced when removing or reducing MIB1 (Supplementary Fig. 11b and c). In addition, we validated WBP2 as NEDD4 target by *in vitro* ubiquitination assay (Supplementary Fig. 11d and e). Taken together, these orthogonal assay and validations support that candidate substrates identified by BioE3 are promising leads for future biological studies.”*

In the volcano plots for the MS data is there any particularly reason why only -log *p*-value was used as a filtering criteria and not fold change as well? Presumably when comparing wild-type vs catalytically inactive or substrate inactive you would be looking for an enrichment (fold change) potential substrate to give higher confidence that the protein hits are real (e.g., >2-fold and *p*-value cut-off).

We agree that in most proteomics experiments, the fold-change enrichment gives higher confidence to the identifications and is an important criterium to consider (usually a fold-change > 2, i.e. log₂ ratio > 1). However, in this study, the enrichment also depends on the amounts of biotinylated bio^{GEF}Ub that modify the substrates, and this is regulated by the particular BirA-E3 fusion that is studied. Thus, substrates that are highly expressed and heavily modified with longer Ub chains, would be more enriched than proteins that are modified with less numbers of biotinylated bio^{GEF}Ub. Therefore, in

order to not miss those low abundant, poorly modified substrates, we decided to not include any filtering regarding the enrichment (log2 ratios), and maintain those proteins (log2 ratio > 0 and <1) as potential substrates of the ligase.

Is there information on the ubiquitin chain linkage information for the Bio^{GEF}Ub that this technique has a preference for? E.g. Bio^{GEF}-Ub preferences for mono- or poly-ubiquitin or is it fairly heterogenous ubiquitin groups that get ubiquitylated – is there a way to compare Bio^{GEF}Ub to overexpressed Ub to see if these ubiquitylate sites or form polyubiquitin linkages differently. As this may help provide information as to which substrates are preferentially tagged, and which are being directed to specific areas of degradation (e.g. K48 and K63) or locations (K11, K23, K27) in the cell. This may help support the notion that some proteins are directed to autophagy (as shown with the MIB1) or the proteasome with RNF4.

In principle, we believe that the use of bio^{GEF}Ub should not alter the specificity of any given E3 ligase to promote K63 versus K48 or other type of linkages Because bio^{GEF} is located at the N-terminus of Ub, linear chain formation may be reduced since incorporation could act as a chain terminator. In fact, the capacity of bioUb to form different Ub linkages was already shown previously by Mass Spectrometry analysis (see Figure 5 of this rebuttal letter; Franco *et al.*, 2011; DOI: 10.1074/mcp.M110.002188), with similar behavior as wild type Ub. The bio^{GEF}Ub construct contains all lysines susceptible of ubiquitination. We included a sentence in this aspect in the revised *Discussion* section.

Added text:

“Since bio^{GEF} modifies the Ub N-terminus, the method might work less efficiently for linear chain-specific E3s; it may incorporate as single, chain-terminating modifier. BioE3 should enable identification of monoubiquitinated and other classes of polyubiquitinated substrates, as bioUb has been described to generate the different types of chains¹⁰.”

Figure 5: Quantification of the branched ubiquitin peptides generated by bioUb, detected by MS according to spectral counts (Figure 3D of Franco *et al.*, 2011; DOI: 10.1074/mcp.M110.002188).

In addition, we checked the total amounts of K48 and K63 chains (using specific antibodies) in the HEK293FT TRIPZ-bio^{GEF}Ub cell lines used in this study, and observed that the total K48- and K63-specific chain amounts were not significantly affected when inducing the expression of bio^{GEF}Ub (Fig. 6 of this rebuttal letter).

Figure 6: The induction of bio^{GEF}Ub expression does not affect the distribution of K48 or K63 chains.

In addition, different E3 ligases are able to build and extend different Ub-chain types, depending on preferred lysine linkage sites. With BioE3 samples, we confirmed the correlation of E3-type to chain-type in the eluates of RNF4 (K48 E3, K48 enrichment in the eluates) and NEDD4^{3M} (predominant K63 E3 ligase, predominant K63 enrichment in the eluates), as observed in the Fig. 7 of this rebuttal letter.

Figure 7: RNF4 and NEDD4^{3M} BioE3 eluates are enriched in Ub K48 and K63 chains, respectively.

Can the BioE3 approach be used to identify ubiquitylation/SUMO site information? i.e. was there any identified in this study that can be added to support its utility for ubiquitylomics/SUMOylomics (not just on the protein level). If so, how does this approach compare to the standard di-Gly IP approach? SUMOylation site information will require a different enzyme to cleave as it does not give the standard GG- motif.

We agree that the identification of specific modification sites on substrates by particular E3s would be very valuable information. We think BioE3 would be compatible with current tools, with some optimization of buffers, digests, conditions. Reducing urea concentrations (to allow trypsin digestion) may lead to precipitation of some insoluble substrates, but remaining soluble substrates could yield suitable branched peptides for the standard di-Gly IP or the more precise UbiSite IP approach. While this could lead to the identification of E3-specific Ub-modified sites, this would require new sample

replicas, more optimizations, and significant antibody/MS costs, beyond the scope of the current study. We included site-identification as a future potential of BioE3 potential in the *Discussion* section.

Added text:

“combining it with subsequent ubiquitination site purification approaches (di-Gly IP, UbiSITE IP), could lead to the identification of Ub sites on substrates carried by specific E3s.”

When assessing all the substrates for the specific E3 ligases used in this work, is there a consensus motif/sequence that the E3's use to bind and ubiquitylate? If you align the sequences amongst the different substrates does it give information on a peptide region that all substrates have in common? E.g. the well-studied B-TRCP E3 ligase has a preference for substrates with a phosphorylated DpSG motif. This may provide information on the mechanistic information and aide in developing TPD strategies such as PROTACs.

When performing BioE3 with RNF4, in fact we could validate that substrates were recognised through their SUMO moieties, as almost all of them are described to be SUMOylated and the eluates were enriched in SUMO peptides (detected by LC-MS and by WB in this study). For the other RING type E3s, the substrate recognition model is rarely known, and could be studied by bioinformatics approaches to identify consensus sites. Since substrate recognition could be not only sequence-based, but structure-based, recent advances in structure prediction (AlphaFold2, AlphaFold Multimer, etc) coupled with site-identification could potentially reveal important motifs. However, this requires an extensive and specialized bioinformatics analysis that goes beyond the scope of this work. We agree that this is an interesting point of study and included this possibility of BioE3 potential in the *Discussion* section.

Added text:

“*With proper bioinformatics analysis, BioE3 could also be used to define consensus sequences and motifs that a particular E3 uses to bind and ubiquitinate the substrate.*”

In the case of NEDD4, substrates are usually recognized through PY motifs, and we checked their presence in the substrates identified in this study. We observed that several of them indeed contain PY motifs (see Table 1 of this rebuttal letter).

Table 1: PY motif content on NEDD4 substrate list identified in this study.

ID protein	Gene name	PY motifs	Mass	N° Lysine res.
Q969T9	WBP2	3	261aa - 28kDa	15
Q9Y2J4	AMOTL2	2	779aa - 86kDa	24
Q15018	ABRAXAS2	1	415aa - 47kDa	18
O43809	NUDT21/CPSF5	1	227aa - 26kDa	14
Q8WUM4	PDCD6IP/ALIX	1	868aa - 96kDa	63
O14828	SCAMP3	1	347aa - 38kDa	10
P45974	USP5/UBP5/ISOT	1	858aa - 96kDa	51
O60762	DPM1	1	260aa - 30kDa	18
P00374	DHFR/DYR	1	187aa - 21kDa	17
Q15233	NONO	1	471aa - 54kDa	27
Q13625	TP53BP2	1	1128aa - 126kDa	69

Can the BioE3 approach be benchmarked against a standard E3 ligase IP-MS approach instead of comparing to existing data? As different mass specs and set ups can change the results – it would be

good to show in house that the BioE3 approach has an advantage or is complementary to a standard E3 IP-MS set up.

Thanks for this question, but we are not sure exactly what the reviewer is asking. If question is still pending, we are happy to answer in future correspondence.

We attempt to answer here. In this rebuttal letter and in the manuscript, we have tried to emphasize the difference between E3-interactors and E3-substrates. “Interactor”-type experiments (pull-downs, co-immunoprecipitations – perhaps what the reviewer refers to as “E3 IP”) and proximity methods using promiscuous biotinylators (BioID-E3, TurboID-E3, APEX2-E3 etc) can identify strong and weak binders to an E3 ligase, which may or may not be substrates for Ub modification by that E3. And with those experiments, there is no way to know whether an interactor is substrate or not. On the other hand, BioE3 depends on the proximity of the BirA-E3, bio^{GEF}Ub, and the substrate(s) to act and achieve labelling, so we believe that it identifies E3-substrates.

As mentioned in other comments above, when studying a single E3 ligase, one might benefit from pursuing multiple approaches (FLAG-E3 IP, endogenous IP, TurboID-E3, and BioE3 [as described here]), all followed by mass spectrometry analysis. One could also vary positioning of the tags (e.g. Nterm vs Cterm FLAG, TurboID, BirA). Resulting datasets could be compared for shared and unique hits to answer questions about how each method performs for isolating interactors vs substrates. Variations between sample prep and mass spec set-ups will always be an issue, and complex experiments with many steps are sure to have variations. While BioE3 can catalog substrates of a particular E3 ligase in a particular cell line, it is far from a definitive set of possible substrates. Performing BioE3 in different cell types with different stimuli, performed by different groups using different transfections/media/capture reagents, analyzed on different MS devices: all this makes the method difficult to “benchmark”. Rather than yielding a definitive “E3-ubiquitinome”, we think BioE3 will be a potent tool for discovery of new unexpected substrates, for understanding how substrate-choice responds to stimuli, and for lending insights into which biological pathways are regulated by a particular E3.

Comments referring to specific figures:

Figure 1: Currently written as:

HEK293FT
U2OS

For clarity, could this please be changed to

HEK293FT/
U2OS

For clarity, we included the change in Fig. 1.

Figure 2

Was BirA alone (without attachment to an E3 ligase) ever used as a background control given the amount of leakiness of expression as observed in some Figures?

Could BirA (inactive) also be used as a negative control?

In Fig.2, we used BirA alone to show the unspecific-labelling of the WT AviTag (bio^{WHE}) in comparison to the low affinity bio^{GEF} tag. BioE3 would not work if BirA is labelling bioUb regardless of E3-fusion.

Thus, when performing BioE3 experiments, we believe that the proper negative control is the fusion of an active BirA to the catalytically-inactive (CA) version of the ligase. This negative control was used in all the experiments. Importantly, in this scenario, the BirA-E3^{CA} negative control localizes identically to the BirA-E3^{WT} and therefore, the differences observed in biotinylation are only due to the ligase activity of the E3. A caveat is that sometimes E3s work as dimers or multimers, and the BirA-E3^{CA} can interact with endogenous E3^{WT} and yield a weak labelling. Using a knockdown or knockout cell line for the E3 of interest might yield higher signal/noise ratios, but we did not try this approach. In our opinion, inactive BirA would not be an effective negative control, since the lack of biotinylation is due to the “dead” BirA enzyme in that case, and not due to the inactivity of the ligase.

Figure 3

3A. The abundance of BirA is not the same between WT and CA or deltaSIM. Please show that the difference in biotinylation is not due to this form of enzyme being at a higher concentration (e.g., densitometry of biotin normalised to BirA* or determine the conditions where the protein quantities are the same). I understand that CA and deltaSIM are meant to be negative controls however, experience with some catalytically inactive enzymes demonstrates that they are not always 100% inactive.

We agree with the Reviewer. low levels of background biotinylation can be observed sometimes when using CA versions of the ligases. This could stem from inactivating mutations that have residual activity (i.e. not always 100% inactive) or, as mentioned above, dimerization with WT counterparts to give reduced, but detectable and specific bio^{GEF}Ub labelling. As suggested by the Reviewer, we quantified the relative signal intensity of the biotin blots, normalized to the BirA levels, and represented the differences in signal including statistical analysis (ANOVA with Tukey correction), see Fig. 8 of this rebuttal letter. The Fig. 3a has been updated to include the novel graph, reference to it has been added in the main text, and source data for Fig. 3a (with quantifications and *p*-values of the ANOVA test) has been included in the source data file.

Added text:

“We posited that biotin-labelled substrates seen with RNF4^{WT} compared to the RNF4^{CA} mutant, especially those that significantly accumulated upon proteasomal inhibition, would constitute the ubiquitinated targets of RNF4 (Fig. 3a, biotin blot and biotin signal quantification graph).”

Just to confirm, we would like to point out that we are not using BirA* (promiscuous BirA or BioID, as used in proximity proteomics and often denoted as BirA* with an asterisk) for any of the experiments shown in this manuscript.

Figure 8: Quantification and statistical analysis of RNF4 BioE3 biotinylation activity. (Note: corresponds to Fig. 3a in revised manuscript).

3C. Please indicate which band or smears of bands represent PML. Could these results be quantitated using ImageJ (with added statistics)?

For clarity, the smear that represents PML was included in the figure. RNF4 BioE3 experiment (+ or – arsenic trioxide or ATO) was repeated 2 additional times, and PML signal was quantified and normalised to the BirA-RNF4 levels from the INPUT fraction. We represented the differences in signal including statistical analysis (ANOVA with Tukey correction) in the Fig. 9 of this rebuttal letter. The Fig. 3c has been updated to include the novel graph, reference to it has been added in the main text, and source data for Fig. 3c (with quantifications and *p*-values of the ANOVA test) has been included in the source data file.

Added text:

“PML was significantly enriched after treating the cells with ATO, compared to RNF4^{CA} or RNF4^{ΔSIM} (Fig. 3c, quantified 3-fold enrichment).”

Figure 9: Quantification and statistical analysis of PML isolation upon RNF4 BioE3 and arsenic trioxide (ATO) treatment. (Note: corresponds to Fig. 3c in revised manuscript).

Figure 4

While ubiquitylation and sumoylation substrates were identified, is there any information on the site information of where the Bio^{GEF}Ub is labelling? Is the lysine or NH₂ on the N-terminus of the potential substrate hindering or competing with biotinylation – will this affect selectivity and site specific information?

When analysing data from BioE3 experiments, we checked biotinylated lysine-containing, Bio^{GEF}-derived peptides, but few were observed. We also checked for di-Gly-containing branched peptides, but too few were detected to identify ubiquitination sites with confidence. This is due to the fact that lysines that are modified by the biotinylated bio^{GEF}Ub are present in only a few peptides from a given substrate, and branched peptides are often bulky and poorly detectable by LC-MS. However, if one is

interested in the identification of the specific Ub sites, we believe that BioE3 could be followed by di-Gly IP or UbiSite IP protocols to enrich for modified peptides and facilitate identification of Ub-modification sites. Text mentioning this has been added to the *Discussion* section.

Added text:

“and combining it with subsequent ubiquitination site purification approaches (di-Gly IP, UbiSITE IP), could lead to the identification of Ub sites on substrates carried by specific E3s.”

On the other hand, the lysine or NH₂ on the N-terminus of the potential substrate cannot compete with biotinylation by BirA, since it specifically biotinylates a single lysine in the AviTag/bioTag (bio^{GEF} or bio^{WHE}). We think the reviewer is referring to “lysine competition”, which could be an issue if the promiscuous biotinylation enzyme BirA* (aka BioID, used for proximity proteomics) was being used. We used wild-type BirA in all of the experiments shown in this manuscript (not the promiscuous BirA*/BioID).

Shouldn't the abundance of the BirA*-ligase also be accounted for in your analysis pipeline?

Thanks for the comment. Again, for clarity, we use wild-type BirA for fusions, not BirA*/BioID, in these experiments.

Considering that we only detect the fraction of the proteome that is ubiquitinated by BirA-E3^{WT} (i.e. the E3 of interest). Therefore, the amount of BirA-E3^{WT} present in the streptavidin pull-down depends on the levels of auto-ubiquitinated E3, that is, the amount of biotinylated bio^{GEF}Ub that modifies the E3 itself. Therefore, the abundance of BirA-E3 detected by LC-MS does not directly correlate to its expression level, but of its “ubiquitinated” status (driven largely by auto-ubiquitination). To check and compare BirA-E3 levels, the INPUT fractions are more representative. The levels of the BirA-E3^{WT} and the BirA-E3^{CA} versions have been checked in all the experiments (depicted by BirA blot in all the experiments). For the most part they were comparable, although MIB1 and NEDD4^{3M} exhibited strong auto-ubiquitination activity. While they are ubiquitinating themselves, we assume they are also ubiquitinating *bona fide* substrates.

Nevertheless, checking and comparing BirA-E3^{WT} levels might be more important when comparing E3 ligases with very similar structural motifs and localizations. One example could be the comparison of two F-box-containing substrate receptors (e.g. FBXW-type; cullin RING ligase components), which only differ by numbers of WD-40 motifs (which mediate substrate recognition). To identify specific substrates of one or the other, matched expression levels would be quite important.

Figure 6

-Because an artificial form of Bio^{GEF}Ubnc, which is uncleavable by Dubs, is expressed as a single copy in these cell lines, it would be valuable to show that some of these substrates can be modified by endogenous (in vivo) or recombinant (in vitro) Ub or SUMO to demonstrate that these are indeed putative substrates. The BioE3 approach is great for discovery but I think demonstrating 1 or 2 substrates that this is highly selective to identify true substrates is needed.

We agree with the Reviewer that using orthogonal approaches to validate the substrates would reinforce the conclusions of the manuscript. Because this was a concern for both reviewers, we copy here our response.

In this revision, we aimed to validate different targets identified for all the E3s used in this study (compiled in new Supplementary Fig. 11).

Using a common tool for enriching ubiquitylated proteins (His⁶-Ub), we performed His⁶-Ub pull-down experiments expressing different RING ligases (BirA-RNF4, MIB1, MARCH5, RNF214 and NEDD4) and could confirm several hits as targets since their ubiquitinated levels were higher using wild-type versions compared to their respective CA versions (see Supp Fig. 11a).

Also, we generated MIB1 *knock out* (MIB1-KO) cells and performed an enrichment of the endogenous ubiquitome using TUBEs (Mattern *et al.*, 2019; DOI: 10.1016/j.tibs.2019.01.011). In parallel, we also performed a TUBEs experiment by *knocking down* MIB1 using siRNAs. Cells were also treated or not with the proteasomal inhibitor bortezomib to further enrich the ubiquitinated fraction. Using these approaches, we could validate CEP131 and USP9X as Ub targets of MIB1, as the endogenously ubiquitinated fractions of those hits were reduced when *knocking out* or *knocking down* MIB1 (see Supp Fig. 11b and c, respectively).

In the case of NEDD4, we also performed an *in-vitro* ubiquitination assay using purified WBP2 as a substrate. We confirmed WBP2 as a direct target of NEDD4, with no ubiquitination observed using the transthiolation mutant NEDD4^{CS} (see Fig. 7d and e of this rebuttal letter). We note that WBP2 was also previously described as NEDD4 substrate (Kim *et al.* 2009; DOI: 10.1128/MCB.00240-09).

In summary, we used diverse methodologies (GFP-trap pulldowns [GFP-TCP1], His⁶-Ub, TUBEs, *in vitro*) to verify the ubiquitination of substrates of all the E3 ligases under study, using diverse negative controls as CA mutants, KOs and KDs. These orthogonal validations have been added as Supplementary Fig. 11. Comments on these validations were included in the main text, *Results* section, and the *Methods* section has been updated.

Added text in Results:

“Orthogonal validations of BioE3 targets

*To further support that targets were indeed ubiquitylated by the different E3 ligases that we studied, we used orthogonal approaches for validation. We performed His⁶-Ub pulldown experiments, comparing wild-type and catalytic mutant versions of the different BirA-E3s used in this study to confirm ubiquitination status of selected hits (e.g. SUMO-conjugates:RNF4; USP9X/CEP131:MIB1;ARFGAP1:MARCH5; ROCK1:RNF214; CCT8/TP53BP2:NEDD4-3M;Supplementary Fig. 11a). In addition, we generated MIB1 knockout (MIB1-KO) cells, or depleted MIB1 using RNAi, and then enriched the endogenous ubiquitome using TUBEs, with and without proteasome inhibition by bortezomib (Mattern *et al.*, 2019; DOI: 10.1016/j.tibs.2019.01.011). Using this approach, we could validate CEP131 and USP9X as Ub targets of MIB1, as the endogenously ubiquitinated fraction of those hits were reduced when removing or reducing MIB1 (Supplementary Fig. 11b and c). In addition, we validated WBP2 as NEDD4 target by *in vitro* ubiquitination assay (Supplementary Fig. 11d and e). Taken together, these orthogonal assay and validations support that candidate substrates identified by BioE3 are promising leads for future biological studies.”*

Figure 7

7F. Please add a lane to the WB figure to show the abundance of each enzyme here that led to these results.

The abundance of each enzyme (shown to be equal, depicted by the BirA blot) has been included in the Fig. 7f of the article (see Fig. 10 of this rebuttal letter), and a comment has been added in the main text.

Added text:

“Importantly, similar and comparable expression levels of BirA-MARCH5^{WT}/MARCH5^{CA} and BirA-RNF214^{WT}/RNF214^{CA} were also validated (Fig. 7f, BirA blots).”

Figure 10: BirA-MARCH5 / BirA-MARCH5^{CA} and BirA-RNF214 / BirA-RNF214^{CA} expression levels are similar for LC-MS experiment (Note: corresponds to Fig. 7f in revised manuscript).

REVIEWERS' COMMENTS

Reviewer #1 (Remarks to the Author):

I have examined the revised manuscript and I don't have any additional concerns.

Reviewer #2 (Remarks to the Author):

Sutherland and colleagues have provided a thorough and comprehensive rebuttal. This includes a series of additional experiments which have addressed our main comments around validating novel ubiquitylation substrates using orthogonal methods.

Overall, I believe the authors have satisfied all of our comments.